# Susceptibility to disease (tropical theileriosis) is associated with differential expression of host genes that possess motifs recognised by a pathogen DNA binding protein

Stephen D. Larcombe[1☯¤], Paul Capewell[1☯], Kirsty Jensen[2], William Weir[3], Jane Kinnaird[1], Elizabeth J. Glass[2], Brian R. Shiels[1] *

1 Institute of Biodiversity, Animal Health, and Comparative Medicine, University of Glasgow, Glasgow, United Kingdom, 2 Division of Infection & Immunity, The Roslin Institute and R(D)SVS, University of Edinburgh, Edinburgh, United Kingdom, 3 School of Veterinary Medicine, University of Glasgow Veterinary School, University of Glasgow, Glasgow, United Kingdom

☯ These authors contributed equally to this work.
¤ Current address: Institute of Immunology and Infection, School of Biological Sciences, University of Edinburgh, Edinburgh, United Kingdom
* brian.shiels@glasgow.ac.uk

## Abstract

### Background

Knowledge of factors that influence the outcome of infection are crucial for determining the risk of severe disease and requires the characterisation of pathogen-host interactions that have evolved to confer variable susceptibility to infection. Cattle infected by *Theileria annulata* show a wide range in disease severity. Native (*Bos indicus*) Sahiwal cattle are tolerant to infection, whereas exotic (*Bos taurus*) Holstein cattle are susceptible to acute disease.

### Methodology/Principal findings

We used RNA-seq to assess whether *Theileria* infected cell lines from Sahiwal cattle display a different transcriptome profile compared to Holstein and screened for altered expression of parasite factors that could generate differences in host cell gene expression.

Significant differences (<0.1 FDR) in the expression level of a large number (2211) of bovine genes were identified, with enrichment of genes associated with Type I IFN, cholesterol biosynthesis, oncogenesis and parasite infection. A screen for parasite factors found limited evidence for differential expression. However, the number and location of DNA motifs bound by the TashAT2 factor (TA20095) were found to differ between the genomes of *B. indicus* vs. *B. taurus*, and divergent motif patterns were identified in infection-associated genes differentially expressed between Sahiwal and Holstein infected cells.

**Data Availability Statement:** RNA-seq data are publicly available at the NCBI GEO online repository under accession numbers GSM4824553–GSM4824563.

**Funding:** This work was funded by BBSRC: grant BB/L004739/1 and a BBSRC Strategic Programme grant (Control of Infectious Diseases: [BB/P013740/1]). The funder had no role in study design, data collection and analysis, decision to publish, or preparation of the manuscript.

**Competing interests:** The authors have declared that no competing interests exist.

## Conclusions/Significance

We conclude that divergent pathogen-host molecular interactions that influence chromatin architecture of the infected cell are a major determinant in the generation of gene expression differences linked to disease susceptibility.

## Introduction

It is well established that the outcome of infection by a pathogen can vary between individuals from asymptomatic to fatal disease. How such variability is generated is of great importance, particularly for emerging viral disease such as COVID-19 and infectious disease of livestock. Adaptation of hosts that display no or mild clinical signs upon pathogen infection is thought to have arisen via competitive co-evolution [1,2]. In this process, mechanisms deployed by the pathogen to disrupt defence mechanisms are countered by the selection of genetic traits that attenuate this disruption. Over time this can lead to establishment of a general relationship where, upon infection, immunopathology is minimised to promote host survival and pathogen transmission. Hosts that have not co-evolved with a pathogen, however, are more susceptible to severe disease pathology when infected. Thus, host breeds that are generally tolerant of infection or susceptible to acute disease provide extremes of a susceptibility spectrum that can be exploited to investigate pathogen-host interactions that give rise to variable infection outcomes.

Tropical theileriosis, caused by the tick-borne apicomplexan parasite *Theileria annulata*, is a disease of great economic significance to livestock producers over large areas of the Old World [3,4]. It impacts both large and smallholding production systems with losses caused by mortality, reduced productivity and costs for treatment and control. *B. taurus* cattle show greater susceptibility to acute tropical theileriosis than *B. indicus* breeds native to regions endemic for *T. annulata* (reviewed in [5]). Susceptibility to the disease constrains the use of *B. taurus* in countries such as India, where a major crossbreeding programme aims to select for productive cattle tolerant to *T. annulata* infection [6]. While this has mitigated losses from acute disease, crossbred animals still show more pronounced signs of subclinical infection relative to native breeds [7,8]. Thus, strategies to improve disease tolerance of productive exotic breeds are needed but require greater understanding of how differential susceptibility to disease is conferred.

Previous work has investigated mechanisms of susceptibility to tropical theileriosis. Experimental challenge confirmed that infected Sahiwal (*B. indicus*) calves showed reduced measures of clinical pathology, relative to susceptible Holstein (*B. taurus*), including lymph node enlargement and fever [9]. A reduced pro-inflammatory response was observed for the Sahiwal calves, indicating that in a tolerant host immune pathology is limited, while in a susceptible one disease is caused by parasite-mediated dysregulation of the host immune response (reviewed in [2]).

During tropical theileriosis, sporozoites inoculated by a feeding tick rapidly invade leukocytes (of predominantly myeloid origin) and differentiate into the intracellular macroschizont stage [10]. Macroschizont formation is accompanied with transformation of the infected leukocyte into a cancer like cell. Infected cells proliferate in lymph nodes and metastasise throughout the lymphoid system to establish foci in the liver and lungs: death can occur via pneumonia like condition [11,12]. Based on the kinetics of disease it has been concluded that the major mechanism of tolerance is linked to the macroschizont infected leukocyte [9].

Moreover, five independent macroschizont infected cell lines derived from Sahiwal showed a reduced ability to transverse Matrigel compared to five Holstein lines, indicating that reduced invasiveness of the infected cell is linked to tolerance of infection [13].

Transformation of the bovine leukocyte by *Theileria* parasites occurs via constitutive activation of bovine transcription factors that act as inflammatory mediators and promote oncogenesis (for example, NF-κB and AP1). Activation is accompanied with an extensive, irreversible reorganisation of the infected cell transcriptome [14] that alters the expression levels of many genes encoding proteins with the potential to promote oncogenesis or disrupt the immune response [15]. Thus, parasite-host interactions that activate and/or moderate inflammatory mediators in the transformed, infected cell are likely to be associated with disease susceptibility.

Only a handful of *Theileria*-encoded factors that translocate to the host cell compartment and modulate the infected leukocyte have been identified. TaPIN is a peptidyl-prolyl isomerase that was reported as secreted into the host cell by transforming *Theileria* species. It contributes to AP1 activation by destabilizing the FBW7 ubiquitin ligase that targets c-Jun, a key component of the AP1 heterodimer, for degradation [16]. Likewise, the Ta9 protein is located in the host cytoplasm and has recently been reported as an activator of AP1 [17]. Lastly members of the *TashAT* gene cluster are only found in transforming *Theileria* genomes, they are translocated to the host nucleus, possess AT hook domains homologous to those of mammalian HMGA proteins, bind DNA and modulate bovine gene expression [18–20]. It is unknown whether any of these candidate factors show altered expression in infected cells from tolerant breeds, but this could significantly influence the infected cell phenotype and potentially the severity of disease it generates.

To investigate whether differences in disease tolerance are mediated by pathogen-host interactions that modulate gene expression in the *Theileria*-transformed leukocyte, we have compared the transcriptome profile of the available set of *ex vivo* infected cell lines derived from Sahiwal *vs*. Holstein calves [13,21]. These low passage lines, although genetically diverse, offer the best infected cell lines available for comparative RNA-seq that relates to infection *in vivo*, since long term passage or cloning of infected cells results in attenuation of pathogenicity and a loss of parasite diversity [13,22]. The current work builds on an earlier microarray analysis, limited to 5000 immune related genes, that demonstrated differences in expression of genes encoding factors that regulate immune cell interaction [23]. The microarray study only covered the response to early infection events (first 72 h after invasion), and only a relatively small number of differentially expressed genes were identified (150 genes), reducing the power of pathway analysis. Moreover, a comparison of the host or parasite transcriptome of the Sahiwal and Holstein cell lines was not performed in the previous studies [13,21]. The results from our analysis show that infected cell lines from cattle breeds differentially susceptible to tropical theileriosis display significant differences in expression of bovine genes associated with parasite infection, innate immunity, cholesterol biosynthesis and oncogenesis. Furthermore, a screen for DNA motifs bound by the parasite encoded factor, TashAT2 identified differences in number and pattern between the genomes of *B. indicus vs. B. taurus*, which included infection-associated genes differentially expressed between Sahiwal and Holstein infected cell lines. These data provide a platform to characterise parasite-host interactions that have evolved to generate the wide range of disease outcomes that occur following infection.

## Methods

### Parasite infected cell lines and culture conditions

We used eleven infected cell lines, previously established *ex vivo* from the peripheral blood of cattle experimentally challenged with *T. annulata*. Six of these were derived from infected

Sahiwal calves (*B. indicus*) and five were derived from infected Holstein calves (*B. taurus*), as reported [13,21]. Both Sahiwal and Holstein animals were infected with sporozoites from the same parasite stock, *T. annulata* Hissar and cultured in RPMI-1640 medium (Sigma-Aldrich) supplemented with 10% FCS, 4 mM L-glutamine and 50 μM β-mercaptoethanol at 37˚C for a limited period (less than 10 passages). All cell lines were fully established based on the level of infection (>95% macroschizont infected cells). Clones were not generated in the original or current studies in order to retain the heterogeneity that naturally occurs upon infection *in vivo*, and because cloning and prolonged passage result in an attenuation of infected cell pathogenicity [13,22]. Thus, RNA was isolated from low passage cell lines, between passages 4–6, for Sahiwal and between passages 6–8, for Holstein. Cells were harvested by centrifugation (120 *g* for 10 min) after 48 h of culture. RNA was isolated using Trizol (Thermo Fisher Scientific) followed by purification using the RNeasy mini kit (Qiagen), including an on-column DNAse digestion step, and using the suppliers' protocols. RNA quality and quantity were then assessed using a Nanodrop ND-1000 spectrophotometer (Thermo Fisher Scientific) and by gel electrophoresis.

## RNA-seq

Between 160 and 260 μg DNAse-treated RNA was sequenced for each sample at the Centre for Genomic Research (CGR), Liverpool. Dual-indexed strand-specific RNA-seq libraries were made from total RNA, using NEBNext Poly(A) mRNA Isolation and Ultra Directional RNA Library Preparation kits (New England Biolabs). Sequencing was performed using one lane of Illumina HiSeq 4000 (Paired-end, 2×150 bp sequencing, generating data from >280 M clusters per lane). Following sequencing, paired end reads were QCed and prepared for analysis at CGR: FASTQ files were trimmed for Illumina adapter sequences using Cutadapt version 1.2.1 [24] using option -O, so the 3' end of any reads which matched the adapter sequence for 3 bp or more were trimmed. The reads were further trimmed using Sickle version 1.200 [25], with a minimum window quality score of 20. Reads shorter than 20 bp after trimming were removed. Approximately 40 million paired end reads were generated from each sample.

## Read mapping, differential expression analyses and pathway analyses

The trimmed FASTQ files for each sample were mapped to the *B. taurus* transcriptome: v UMD3.1 downloaded from Ensembl Biomart [26]. This strategy was used because of the greater fidelity of the *B. taurus* reference genome sequence relative to *B. indicus*, and on the premise that the two subspecies share a high level of sequence identity in mRNA coding regions of the genome. Bowtie2 [27] was used to map reads and generate SAM files, and the Python script "SAM2counts" [28] to generate reference sequence counts for each transcript. We collapsed reads mapping to multiple transcripts to the same Ensembl gene ID prior to analysis. Differential Expression analyses were performed on the count files using DESeq2 [29], to compare the five *B. taurus* samples to the six *B. indicus* samples. The *B. taurus* data values were used as the primary comparator, such that negative fold changes in gene expression are designated as lower expression in *B. taurus vs. B. indicus*, and positive fold changes are designated as higher in *B. taurus vs. B. indicus*. Genes with a false discovery rate (FDR) less than 0.1 (10%) were considered significantly differentially expressed between *B. indicus* relative to *B. taurus* infected cells lines. Following differential expression analyses, we used Ensembl Biomart to annotate the read files and Ingenuity Pathway Analysis (IPA) (Ingenuity Systems Inc., Redwood City, USA) to elucidate significantly modulated gene networks, disease pathways and potential effector molecules. We also mapped the RNA-seq reads to the *T. annulata*

genome [30], to assess whether parasite encoded genes are differentially expressed between the established macroschizont infected *B. taurus vs. B. indicus* cell lines.

## Comparison with TBL20 infection associated dataset

To elucidate mechanisms that might underpin differences in breed susceptibility to *T. annulata* infection, we matched our data set of breed-associated differentially expressed genes with a data set of host cell genes whose differential expression was identified as associated with infection by *T. annulata*. The infection-associated data set was obtained by comparison of the transcriptome (obtained by microarray analysis) of uninfected (BL20) *vs.* macroschizont infected (TBL20) bovine lymphosarcoma cell lines, treated and not treated with the anti-*Theileria* drug buparvaquone. Full details can be found in [14,15]. As the two datasets used different systematic ID codes (Ensembl *vs.* Entrezgene), we first used Ensembl Biomart to filter the microarray BL20/TBL20 dataset by finding common Gene stable IDs. Since some genes were not found in this way we also used DAVID [31] and PANTHER [32] to generate alternative gene symbols to screen for overlap between gene lists independent of common gene IDs. To check for statistical significance, the representation factor was calculated as the number of genes in common ÷ expected number of genes; where the expected number of genes is given by: (number of genes in group 1 × number of genes in group 2) ÷ number of genes in total. In our case we used 19,018 as the total number of genes as this is the number of unique identifiable bovine genes included on the microarray. The p value was then computed as a normal approximation of the exact hypergeometric probability [33].

## Identification of infected host cell genes encoding predicted secreted or receptor proteins

Ensembl Biomart was used to generate the peptide sequences for all bovine genes classed as differentially expressed. A pipeline of publicly available resources was then used to perform *in silico* predictions. Signal P version 5.0 [34] generated a list of proteins with a predicted signal peptide (cutoff of >0.5 probability). Sequences for this new list were then screened for the presence of transmembrane domains or GPI anchors. This was performed using TMHMMserver v 2.0 [35] for transmembrane domains, and PredGPI [36] for GPI anchors. Any predicted proteins that scored positive for either motif was excluded from the list. The final list of proteins was obtained by screening against subcellular location prediction and gene ontology (GO) data available in the Genecards database version 4.14 [37]. Only proteins with a top confidence score as extracellular were retained in the final list. To generate a list of receptor proteins, we used gene ontology (GO) software to look for evidence of annotation as a receptor, or receptor-like domains in our data sets of differentially expressed genes. Thus, lists of annotations, protein family and classes were generated using Ensembl, DAVID, PANTHER and IPA. The data from each resource was collated, and we excluded any molecule that was not a confirmed or predicted receptor. Following receptor classification, the resulting gene list was curated by eye and nuclear receptors excluded, as we were primary interested in a cell surface location.

## Validation of RNA-seq results by qRT-PCR

Validation of differential gene expression was carried out by qRT PCR, using RNA from the eleven *T. annulata* cell lines isolated on a different day than the samples used for RNA-seq. Primer pairs specific for 20 genes were designed (S1 Table) and qRT-PCR performed, as described previously [15]. Briefly, 500 ng of total RNA from each sample was used to synthesise cDNA, using the Affinity Script cDNA Synthesis Kit (Agilent Technologies) and Oligo-dT

as primer. 1 μl cDNA for each sample was then used for qRT-PCR, using the Brilliant III Ultra-fast SYBR®Green qPCR Master mix (Agilent technologies) and the Stratagene Mx3005P system. Comparative quantitative analysis of gene expression across samples was performed using Stratagene MxPro Software. *REPS1* (RALBP1 associated Eps domain containing 1) was utilised as a house keeping gene, based on previous identification of constitutive expression in *Mycobacterium bovis* infected macrophage cells [38] and initial validation using RNA from the eleven *T. annulata* infected cell lines. The ΔΔCT method was then used to calculate fold-change (*vs*. the lowest expressed sample). The data were transformed on the $\log_{10}$ scale to stabilize the variance and the mean group differences (Sahiwal *vs*. Holstein) were tested using Student's t tests.

## Bioinformatic screen for DNA motifs bound by TashAT2

A consensus of AT rich motifs preferentially bound by the AT hook domain of recombinant TashAT2 was generated using previously published data [19]. This was used to generate a PROSITE pattern (TAAAT(1)N(4,6)T(1)A(3,4)T.) to search the *B. taurus* (UCD1.2) and *B. indicus* (Bos_indicus_1.0) genomes using FUZZNUC [39]. Bedtools [40] was used to find the intersections of identified motifs and introns and exons in the bovine genomes. Proximal upstream and downstream sequences in validated mRNA sequences not characterized as exons or introns were defined as untranslated regions (UTRs) and the intersection of motifs assessed. Sites without intersection were characterised as non-coding. Pathway enrichment of genes with TashAT2 binding motifs was assessed using PANTHER [32] and summary figures generated using CIRCOS [41].

## Immunoblotting and immunofluorescence

Three of the independent Sahiwal infected cell lines (SA-C) and three Holstein lines (HA-C) were randomly selected, passaged 1X, cultured for 48 h and harvested, as described above. Cells were then washed 3X in phosphate buffered saline (PBS), resuspended in 20 μL of PBS per $10^6$ cells and an equal volume of 2X SDS-sample buffer added. Immunoblotting was performed using standard methodology, as described [42]. Antisera raised against the TA06460 fusion protein (the endoplasmic reticulum form of HSP90 (ER-HSP90)) was used at a dilution of 1:1500, as previously described [42]. For the Ta9 (TA15705) antigen, rat antisera were raised against a 35.7kDa fusion protein [43]. Antiserum production was carried out by the PTU/BS Unit at the Scottish National Blood Transfusion Service, Pentlands Science Park Edinburgh. The anti-Ta9 serum was used at a dilution of 1 in 1200.

Slide preparation, fixation and immunofluorescence were performed as described [15]. The cell lines used were the same as for immunoblotting and the EL24 antiserum specific for TashAT2 [44] was used at 1 in 250. Images were acquired with an Olympus BX60 microscope, SPOT camera and SPOTTM Advanced image software Version Mac: 4.6.1.26, using the matched exposure feature (Diagnostic Instruments).

## Ethics statement

This study did not require ethical approval since it was conducted on *in vitro* cell lines established *ex vivo* from animals in a previous study where ethical approval was met (details available in [23]) and no further human or animal participation was needed.

## Results

### Early passage Holstein and Sahiwal infected cell lines display significant differences in their host transcriptome profile

Six Sahiwal and five Holstein lines infected with *T. annulata* (Hissar) were utilised for differential transcriptome analysis. As reported previously [9], under standard *in vitro* culture conditions, no significant difference in growth potential between the Sahiwal and Holstein lines was observed. RNA was isolated from each line and RNA-seq performed: the RNA-seq data are publicly available at the NCBI GEO online repository under accession numbers GSM4824553–GSM4824563. The results obtained indicate pronounced differences in host gene expression levels between *T. annulata* infected cells derived from Holstein or Sahiwal cattle. These differences were marked in terms of the large number of genes differentially expressed: 2211 genes (without duplicates) at FDR < 0.1 (adjusted), subsequently referred to as Holstein/Sahiwal differentially expressed (DE) dataset (H/S-DE). The full list of H/S-DE genes can be viewed in S1 File. Of the 2211 DE genes there was an almost even split in genes displaying modulated expression associated with one breed relative to the other; thus, 1068 genes (48.3%) were expressed at lower levels in the Holstein infected lines compared to Sahiwal. Furthermore, analysis of the proportion of reads mapped to the *B. taurus* genome showed no statistical difference between the 6 Sahiwal relative to the 5 Holstein RNA-seq sample sets (S2 Table). These results support the premise that sequence identity between mRNA coding regions of the two genomes allow comparative analysis of gene expression using the *B. taurus* genome as the reference.

A demonstration that the H/S-DE data set represents differences in gene expression profiles between infected cells from Holstein *vs*. Sahiwal was provided by principal component analysis (PCA). Clustering of the samples with host type can clearly be seen in the plot shown in Fig 1, where PC1 explained 35% of variation in the data, although a sizeable degree of variability

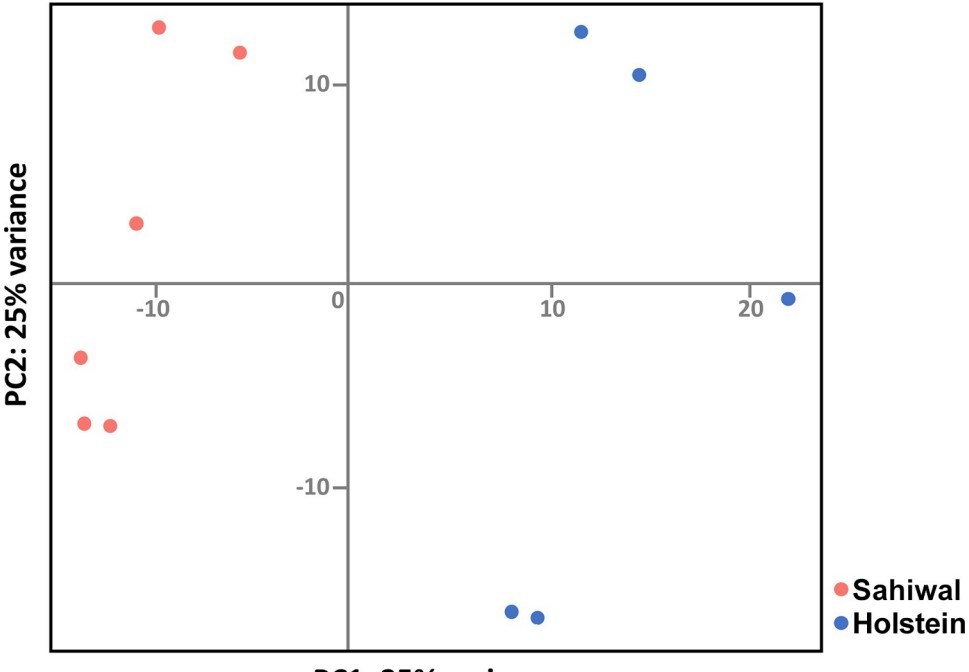

**Fig 1. Clustering of samples by host breed.** A PCA plot showing clustering based on the H/S-DE data set derived from five samples of Holstein (blue) or six samples of Sahiwal (red) cell lines infected with the *T. annulata* macroschizont.

within each breed, that was more pronounced for the Holstein samples, was also detected (PC2 displaying 25% variance).

Indicative of the clear separation between the two breeds, a substantial proportion of genes in the data set showed a consistent difference between the breed types, with 439 of the 500 most statistically significant H/S-DE genes displaying differential expression across all Sahiwal *vs.* Holstein samples. Specific examples include the IFN stimulated gene (ISG) *MX1* expressed at lower levels in all Holstein samples relative to Sahiwal cells (-4.62, $\log_2$); whereas the FAT1 cadherin encoding gene, associated with TGFB signalling, was consistently expressed at a higher level for Holstein cell lines (7.49, $\log_2$; S1 File).

Not unexpectedly, from the PCA, a sizeable number of genes indicated as differentially expressed between the breeds did not display a difference that was consistent across all six Sahiwal *vs.* five Holstein samples. A good example being the gene encoding pro-inflammatory cytokine IL-6, designated as significantly higher in Holstein (1.34 $\log_2$), where two Holstein samples showed lower expression values than the two highest values recorded for the Sahiwal cell lines (S1 File). Thus, genes of this type are indicated as generally differentially expressed between the two breeds but show a more substantial level of variable expression within a breed, and as reported previously [21] this appeared to be more marked for the Holstein sample set.

Table 1 shows the top 19 genes displaying greatest higher or lower expression (fold change $\log_2$) in Holstein (*B. taurus*) relative to Sahiwal (*B. indicus*) infected leukocytes. A number of these genes possess gene ontology (GO) annotation indicating they could play a role in determining phenotypic differences between Holstein and Sahiwal macroschizont infected cells. Thus, for genes with higher relative levels of expression in Holstein infected cells: Slit Guidance Ligand 2 (*SLIT2*), functions in cancer and leukocyte chemotaxis/infiltration [45,46]; Ubiquitin D (*UBD/FAT10*), operates in protein modification/degradation and activation of pro-inflammatory mediators (reviewed in [47]): *GULP1*, is involved in phagocytosis [48] and associated with emphysema [49], a clinical symptom of acute theileriosis [50]. For the genes designated as expressed lower in Holstein infected cells relative to Sahiwal, those encoding proteins associated with a tumour suppressor/inhibitor of metastasis function (e.g. *ANXA8*; *PAX6*; *COL4A1* and *ADAMTS18*) or associated with the innate immune/ISG response were of particular interest. Of the 68 IFN type I response genes [51] identified in the H/S-DE data set, 56 displayed lower expression in Holstein derived infected cells relative to Sahiwal (S1 File).

## IPA reveals breed-associated modulation of gene expression linked to innate immunity, cholesterol biosynthesis and oncogenesis

Differential susceptibility to theileriosis could arise via pre-existing breed associated differences in gene expression between infected (and uninfected) cells or only become manifest after infection of the bovine leukocyte. Therefore, we applied Ingenuity Pathway Analysis (IPA) on both the full data set of 2211 H/S-DE genes and a more limited data set of Infection Associated (IA) genes. To identify breed associated differences in gene expression that are induced by infection, we overlapped the H/S-DE data set with a set of bovine genes demonstrated to display altered expression in *T. annulata*, macroschizont infected *B. taurus* derived lymphosarcoma (TBL20) cells relative to uninfected (BL20) cells [14]. Of the 2211 H/S-DE genes, 517 overlapped and were indicated as the infection associated Holstein/Sahiwal data set (IA-H/S): see S2 File for the full data set. The representation factor for the 517 genes was 1.4 and the p value was < 4.2 e-15; indicating statistical significance for the obtained overlap (Fig 2). Strikingly the most significant canonical pathways identified by IPA for both the full (S3 File) and infection associated data set (S4 File) were virtually identical, with involvement in innate immunity, cholesterol biosynthesis and oncogenesis highlighted.

**Table 1. Top 19 (by Log₂ fold change) down- and up-regulated genes in *B. taurus vs. B. indicus T. annulata* infected cell lines.**

| Gene | log₂ FC | lfcSE | Padj | Gene symbol | Gene description |
|---|---|---|---|---|---|
| ENSBTAG00000021565 | -26.131 | 3.038 | <0.001 | PRSS2 | serine protease 2 |
| ENSBTAG00000002335 | -11.108 | 1.021 | <0.001 | HAND1 | heart and neural crest derivatives expressed 1 |
| ENSBTAG00000012849 | -9.151 | 1.622 | <0.001 | COL4A1 | collagen type IV alpha 1 chain |
| ENSBTAG00000009842 | -8.637 | 1.417 | <0.001 | CRYM | crystallin mu |
| ENSBTAG00000031355 | -8.014 | 0.780 | <0.001 | LOC529196 | C-C chemokine receptor type 1-like |
| ENSBTAG00000046377 | -7.793 | 0.992 | <0.001 | LOC100336807 | |
| ENSBTAG00000001010 | -7.450 | 1.462 | <0.001 | ADAMTS18 | ADAM metallopeptidase with thrombospondin type 1 motif 18 |
| ENSBTAG00000046412 | -7.448 | 1.056 | <0.001 | LOC512617 | acyl-CoA dehydrogenase family member 10 |
| ENSBTAG00000004561 | -6.888 | 1.114 | <0.001 | PAX6 | paired box 6 |
| ENSBTAG00000038064 | -6.865 | 0.457 | <0.001 | LOC614531 | ras-related GTP-binding protein A |
| ENSBTAG00000005857 | -6.751 | 1.312 | <0.001 | SLC6A1 | solute carrier family 6 member 1 |
| ENSBTAG00000027782 | -6.529 | 0.998 | <0.001 | OR5K1 | olfactory receptor, family 5, subfamily K, member 1 |
| ENSBTAG00000008129 | -6.455 | 1.668 | 0.002 | CLSTN3 | calsyntenin 3 |
| ENSBTAG00000047616 | -6.311 | 1.719 | 0.005 | ZNF114 | zinc finger protein 114 |
| ENSBTAG00000018499 | -5.923 | 1.758 | 0.011 | ANXA8L1 | annexin A8-like 1 |
| ENSBTAG00000035110 | -5.918 | 2.049 | 0.038 | DEUP1 | deuterosome assembly protein 1 |
| ENSBTAG00000019428 | -5.765 | 1.473 | 0.002 | CCR1 | chemokine (C-C motif) receptor 1 |
| ENSBTAG00000018367 | -5.733 | 1.347 | <0.001 | CD6 | CD6 molecule |
| ENSBTAG00000014628 | -5.624 | 0.852 | <0.001 | OAS2 | 2'-5'-oligoadenylate synthetase 2 |
| ENSBTAG00000018694 | 8.087 | 1.629 | <0.001 | ACSS3 | acyl-CoA synthetase short chain family member 3 |
| ENSBTAG00000044185 | 7.729 | 3.016 | 0.075 | SOX6 | SRY-box 6 |
| ENSBTAG00000005108 | 7.513 | 1.576 | <0.001 | SLIT2 | slit guidance ligand 2 |
| ENSBTAG00000025803 | 7.506 | 1.809 | <0.001 | INSYN1 | inhibitory synaptic factor 1 |
| ENSBTAG00000020657 | 7.490 | 0.865 | <0.001 | FAT1 | FAT atypical cadherin 1 |
| ENSBTAG00000003345 | 7.149 | 1.769 | 0.001 | FAT4 | FAT atypical cadherin 4 |
| ENSBTAG00000037649 | 7.111 | 1.212 | <0.001 | VIPR2 | vasoactive intestinal peptide receptor 2 |
| ENSBTAG00000015905 | 7.013 | 1.136 | <0.001 | ARHGAP32 | Rho GTPase activating protein 32 |
| ENSBTAG00000025398 | 6.927 | 1.707 | 0.001 | LOC504548 | ubiquitin D |
| ENSBTAG00000004510 | 6.777 | 1.310 | <0.001 | SARDH | sarcosine dehydrogenase |
| ENSBTAG00000047336 | 6.726 | 2.294 | 0.034 | LOC100847115 | thyrotropin-releasing hormone receptor-like |
| ENSBTAG00000031802 | 6.597 | 1.335 | <0.001 | SPATA16 | spermatogenesis associated 16 |
| ENSBTAG00000014354 | 6.519 | 1.246 | <0.001 | FXYD6 | FXYD domain containing ion transport regulator 6 |
| ENSBTAG00000019340 | 6.436 | 1.438 | <0.001 | PCDH9 | protocadherin 9 |
| ENSBTAG00000021526 | 6.296 | 1.994 | 0.020 | RPRM | reprimo, TP53 dependent G2 arrest mediator homolog |
| ENSBTAG00000027246 | 6.247 | 1.388 | <0.001 | UBD | ubiquitin D |
| ENSBTAG00000020385 | 6.216 | 1.408 | <0.001 | ALDH1B1 | aldehyde dehydrogenase 1 family member B1 |
| ENSBTAG00000016525 | 6.179 | 1.898 | 0.015 | ITGA1 | integrin subunit alpha 1 |
| ENSBTAG00000007141 | 6.159 | 1.438 | <0.001 | GULP1 | GULP PTB domain containing engulfment adaptor 1 |

## Innate immunity pathways

Enriched pathways linked to innate immunity in the infection-associated data set were "Activation of Interferon by Cytosolic Pattern Recognition Receptors" and "Interferon signalling" and these pathways were also highly enriched by the analysis of the full H/S-DE data set. Other related significantly enriched pathways included "Role of RIG1 Like Receptors in Antiviral Innate Immunity" and, "NF-κB Activation by Viruses" (full data set only).

IPA indicates the activation state of canonical pathways via the calculated Z-score, a negative Z score predicting lower activation potential, while a positive score indicates higher

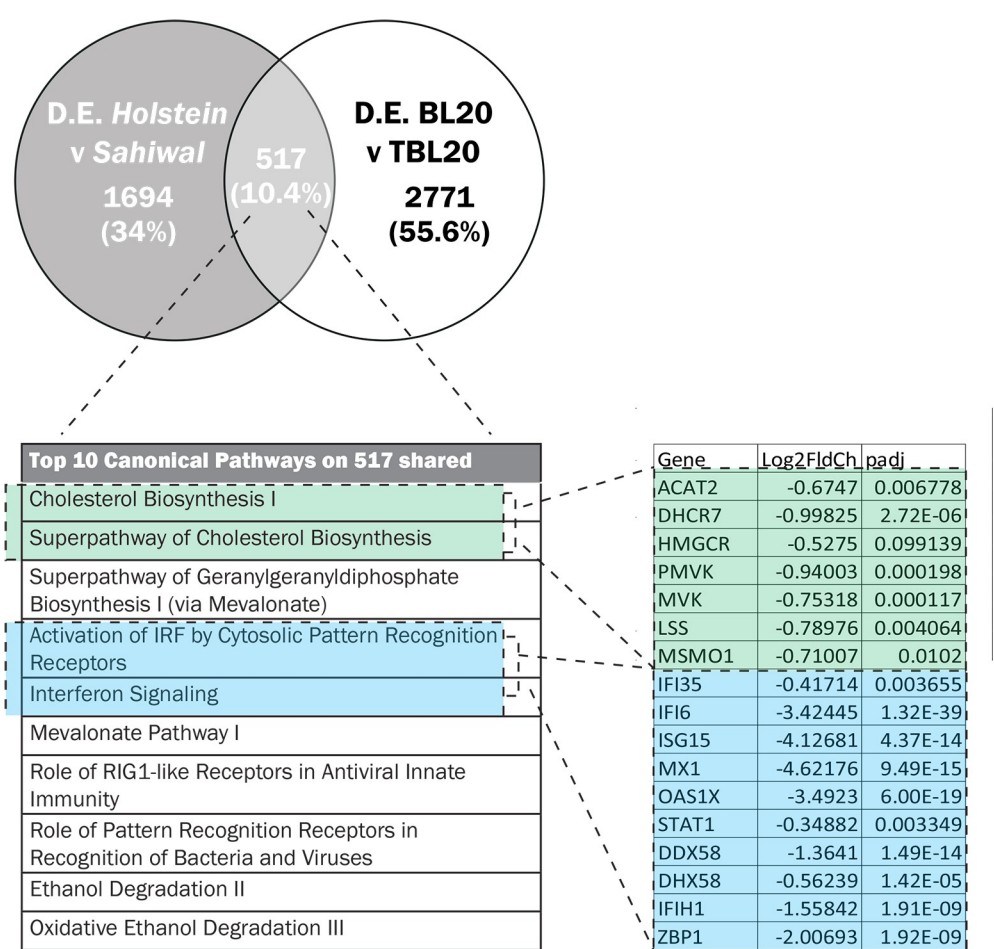

**Fig 2. Overlap of genes modulated between Holstein and Sahiwal breeds of infected cell and *Theileria* infection of BL20 cells.** (A) Venn diagram of overlapping genes, (B) the top IPA canonical pathways and (C) the modulation of infected-associated genes found in enriched pathways linked to cholesterol biosynthesis or the IFN response.

potential. For pathways linked to innate immunity highlighted above, all except NF-κB showed lower activation status in Holstein relative to Sahiwal infected cells (Z score range > -1.3 to -2.8). This finding is illustrated for "Activation of Interferon by Cytosolic Pattern Recognition Receptors" (Fig 3) and "Interferon Signalling Pathways" (Fig 4) generated by IPA for the H/S-DE data set. As would be expected, these two pathways overlap, with lower transcript levels in Holstein infected cells for STAT1, STAT2, IRF9 and the ISG targets of their transcription factor complex. Furthermore, all identified infection-associated ISG genes (*MX1*, *MX2*, *OAS1Y/Z*, *OAS2*, *ISG15*, *IFI6*, *IFI44* and *RSAD2*) showed lower expression values in Holstein infected cells relative to Sahiwal (Fig 2; S2 File).

Nodes, colours, shapes and lines in pathway are as described in legend to Fig 3.

IPA for disease or molecular function also predicted lower innate immune function for Holstein infected cells. As shown in Table 2, the three most significant disease or functions identified for the full H/S-DE data set that are predicted to be lower in Holstein compared to Sahiwal infected cells were "Antiviral response"; "Immune response of cells" and "Antimicrobial response". Furthermore, the IPA function that predicts activation or inhibition of upstream regulators identified inhibition of cytokines, transcription factors and compounds

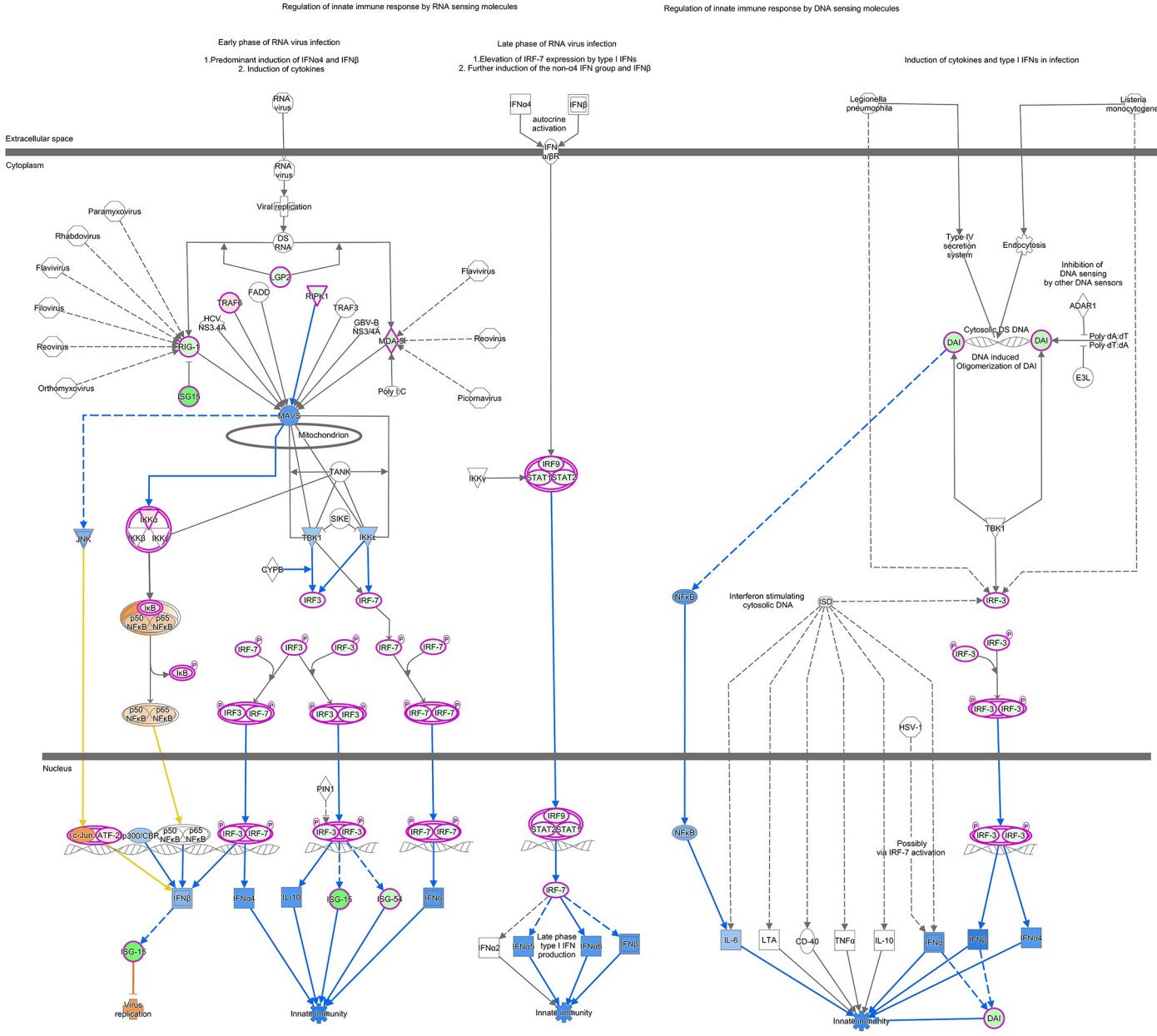

**Fig 3. IPA canonical pathway, "Activation of IRF by Cytosolic Pattern Recognition Receptors".** The nodes represent genes/molecules/complexes in a pathway, and the lines and arrows between nodes indicate known relationships from the Ingenuity Knowledge Base. Nodes with purple outline indicate molecules that were measured as differentially expressed in our dataset, with the intensity of coloured infill indicating the level of up (red) or down (green) regulation of Holstein relative to Sahiwal. The blue- and orange-coloured molecules and lines are predicted activation states generated by the Molecular Activity Predictor function in Ingenuity Pathway Analysis (IPA). Blue colour indicates a predicted inhibition, and orange a predicted activation state in Holstein relative to Sahiwal. Yellow lines indicate relationships where our findings are inconsistent with the state of the downstream molecule. Broad lines with explanatory text beside the pathway indicate the cellular location of molecules in the pathway. The molecules in the pathway are given shapes that indicate their functional class (Nested Circle/Square = Group/Complex, Horizontal ellipse = Transcriptional Regulator, Vertical Ellipse = transmembrane receptor, Vertical Rhombus = enzyme, Square = Cytokine/Growth Factor, Triangle = Kinase, Vertical Ellipse = Transmembrane Receptor, Circle = other). The edges between molecules are also differentiated to indicate the type of relationship between them. Solid lines are direct relationships and dashed lines are indirect.

associated with the interferon response in Holstein relative to Sahiwal infected cells (negative Z scores >2). Very similar results were obtained for both the full and IA data sets (Table 3 & S3 Table).

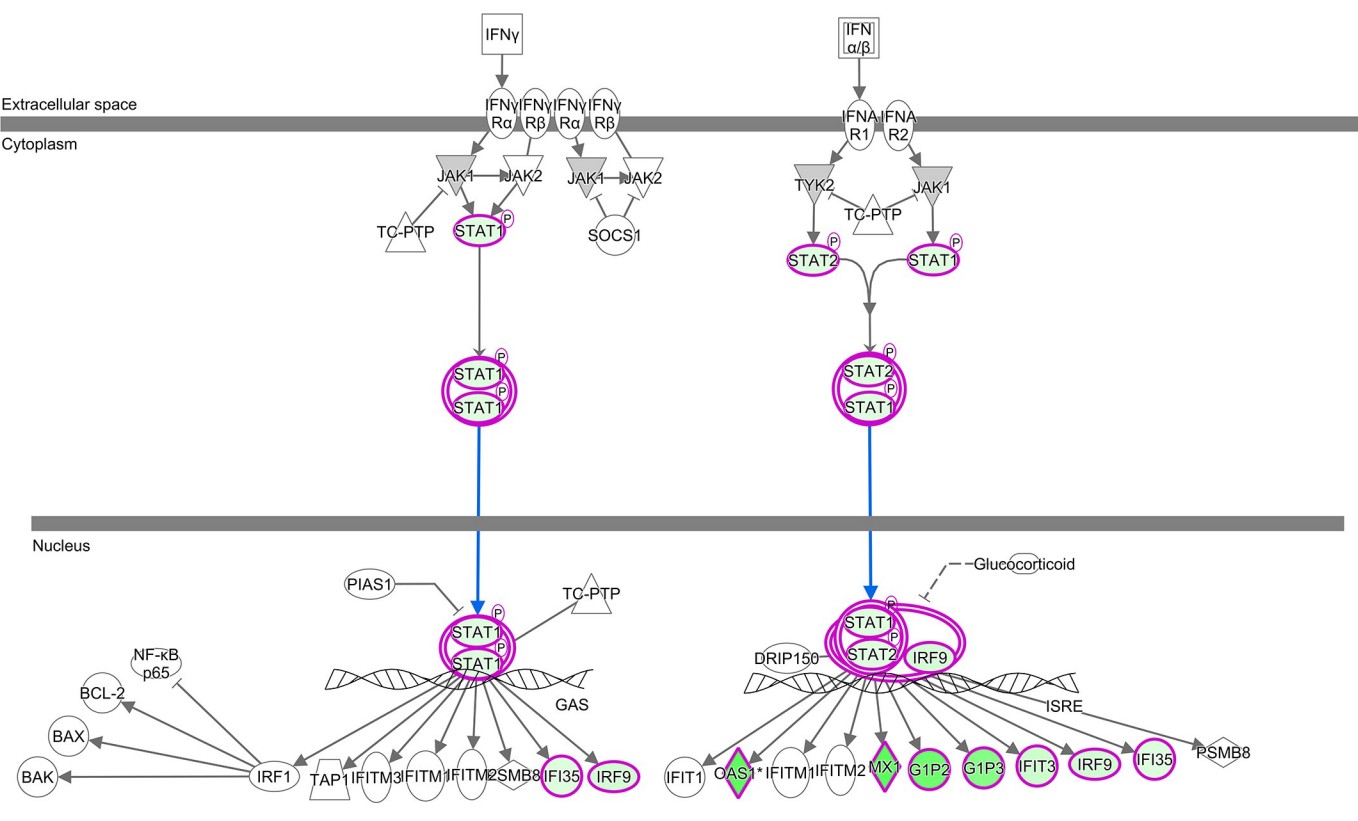

**Fig 4. IPA canonical pathway, "Interferon signalling".**

## Cholesterol biosynthesis

The top three scoring IPA canonical pathways for the infection-associated data set of 517 genes (Fig 2, S4 File) were all linked to cholesterol biosynthesis, with negative Z scores or reduced relative expression values for pathway genes in Holstein infected cells.

Similar results were obtained using the full H/S-DE data set with "Superpathway of Cholesterol Biosynthesis" ranked as the second top canonical pathway and the highest negative Z score recorded (S3 File). Inhibition of the SREBF2 transcription factor, an important positive regulator of genes involved in cholesterol biosynthesis [52], was also predicted (Z score -3.48, P value 6.66 E-06), and relative *SREBF2* expression levels were lower for Holstein (*B. taurus)* infected cells (S1 File). Elevated expression of genes involved in cholesterol biosynthesis is associated with oncogenesis/metastasis via the PI3K/AKT pathway [53,54], which is activated in *Theileria* infected cells [55] and enriched by IPA in both the full H/S-DE and IA data sets (S3 and S4 Files). Therefore, prediction of elevated PI3K/AKT signalling together with lower expression of cholesterol biosynthesis pathway genes in Holstein infected cells was unexpected. However, in other studies decreased expression of cholesterol biosynthesis genes has been associated with activation of Ras/Erk signalling and a more malignant cancer cell phenotype [56].

## Oncogenesis

IPA showed a clear association of the full Holstein *vs.* Sahiwal data set of DE genes with oncogenesis/neoplasia. Thus, the canonical pathways "Retinoic Acid Mediated Apoptosis

**Table 2. IPA for disease pathways and functions enriched in data set of differentially expressed genes between Holstein and Sahiwal *T. annulata* infected cell lines.**

| Category | Diseases or Functions Annotation | p-value | Predicted Activation State | Activation z-score | # Molecules |
|---|---|---|---|---|---|
| Antimicrobial Response, Inflammatory Response | Antiviral response | <0.001 | Decreased | -3.341 | 48 |
| Inflammatory Response | Immune response of cells | <0.001 | Decreased | -2.725 | 84 |
| Antimicrobial Response, Inflammatory Response | Antimicrobial response | <0.001 | Decreased | -3.107 | 56 |
| Neurological Disease | Progressive neurological disorder | <0.001 | Decreased | -2.157 | 133 |
| Cancer, Organismal Injury and Abnormalities, Renal and Urological Disease | Urinary tract cancer | <0.001 | Decreased | -2.236 | 240 |
| Cancer, Organismal Injury and Abnormalities, Renal and Urological Disease | Renal cancer | <0.001 | Decreased | -2.236 | 150 |
| Cell Death and Survival | Cell death of lymphoma cell lines | <0.001 | Decreased | -3.104 | 36 |
| Lipid Metabolism, Small Molecule Biochemistry, Vitamin and Mineral Metabolism | Synthesis of cholesterol | <0.001 | Decreased | -2 | 18 |
| Cell Death and Survival | Apoptosis of lymphoma cell lines | <0.001 | Decreased | -2.81 | 29 |
| Cell Death and Survival | Cell death of macrophage cancer cell lines | <0.001 | Decreased | -2.735 | 11 |
| Cell Death and Survival | Cell death of connective tissue cells | <0.001 | Decreased | -2.034 | 75 |
| Cell-To-Cell Signalling and Interaction, Inflammatory Response | Response of phagocytes | 0.002 | Decreased | -2.498 | 32 |
| Cell-To-Cell Signalling and Interaction, Embryonic Development | Response of embryonic cell lines | 0.002 | Decreased | -3.111 | 12 |
| Cell-To-Cell Signalling and Interaction | Response of myeloid cells | 0.003 | Decreased | -2.466 | 31 |
| Cell-To-Cell Signalling and Interaction, Inflammatory Response | Immune response of phagocytes | 0.003 | Decreased | -2.787 | 29 |
| Embryonic Development, Organismal Development | Development of body trunk | <0.001 | Increased | 3.647 | 153 |
| Infectious Diseases | Viral Infection | <0.001 | Increased | 3.15 | 220 |
| Infectious Diseases | Infection by RNA virus | <0.001 | Increased | 3.075 | 114 |
| Cancer, Organismal Injury and Abnormalities | Development of malignant tumour | <0.001 | Increased | 2.984 | 786 |
| Infectious Diseases | Infection of Mammalia | 0.001 | Increased | 2.978 | 42 |
| Cellular Movement | Cell movement of melanoma cell lines | 0.002 | Increased | 2.715 | 21 |
| Infectious Diseases | Replication of Herpesviridae | <0.001 | Increased | 2.449 | 15 |
| Cancer, Organismal Injury and Abnormalities | Incidence of tumour | <0.001 | Increased | 2.428 | 815 |
| Infectious Diseases | Infection of cells | <0.001 | Increased | 2.331 | 97 |
| Cancer, Organismal Injury and Abnormalities | Malignant solid tumour | <0.001 | Increased | 2.277 | 1292 |
| Infectious Diseases | Replication of Murine herpesvirus 4 | 0.003 | Increased | 2.236 | 5 |
| Cardiovascular System Development and Function, Embryonic Development, Organ Development, Organismal Development, Tissue Development | Cardiogenesis | <0.001 | Increased | 2.198 | 79 |
| Cancer, Organismal Injury and Abnormalities, Reproductive System Disease | Tumorigenesis of reproductive tract | <0.001 | Increased | 2.19 | 459 |
| Cancer, Organismal Injury and Abnormalities, Reproductive System Disease | Female genital neoplasm | <0.001 | Increased | 2.19 | 459 |
| Cancer, Endocrine System Disorders, Organismal Injury and Abnormalities, Reproductive System Disease | Ovarian tumour | <0.001 | Increased | 2.19 | 205 |
| Cancer, Endocrine System Disorders, Organismal Injury and Abnormalities | Endocrine gland tumour | <0.001 | Increased | 2.183 | 1024 |
| Cancer, Gastrointestinal Disease, Organismal Injury and Abnormalities | Digestive organ tumour | <0.001 | Increased | 2.161 | 1130 |
| Cancer, Organismal Injury and Abnormalities | Frequency of tumour | <0.001 | Increased | 2.152 | 801 |
| Cancer, Haematological Disease, Organismal Injury and Abnormalities | Hematologic cancer | 0.002 | Increased | 2.088 | 329 |
| Cancer, Gastrointestinal Disease, Hepatic System Disease, Organismal Injury and Abnormalities | Liver tumour | <0.001 | Increased | 2.017 | 556 |

**Table 3. Top activated or repressed upstream regulators predicted by IPA from gene targets within the H/S-DE data set.**

| Upstream Regulator | Molecule Type | Predicted Activation State | Activation z-score | p-value of overlap | Genes in dataset (Number of regulators from data in network) |
|---|---|---|---|---|---|
| IFNA2 | cytokine | Inhibited | -5.284 | <0.001 | 108 (11) |
| IRF7 | transcription regulator | Inhibited | -5.147 | <0.001 | 137 (13) |
| Interferon alpha | group | Inhibited | -5.024 | <0.001 | 180 (13) |
| IRF3 | transcription regulator | Inhibited | -4.863 | <0.001 | 120 (12) |
| PRL | cytokine | Inhibited | -4.788 | <0.001 | 241 (10) |
| poly rI:rC-RNA | biologic drug | Inhibited | -4.586 | <0.001 | 292 (13) |
| IFNL1 | cytokine | Inhibited | -4.577 | <0.001 | 374 (15) |
| IFN Beta | group | Inhibited | -4.506 | <0.001 | 233 (11) |
| STAT1 | transcription regulator | Inhibited | -4.412 | <0.001 | 318 (12) |
| Ifnar | group | Inhibited | -4.245 | <0.001 | 181 (14) |
| IFNG | cytokine | Inhibited | -4.078 | <0.001 | 367 (14) |
| IRF1 | transcription regulator | Inhibited | -3.994 | <0.001 | 317 (13) |
| CpG ODN 2006 | chemical reagent | Inhibited | -3.846 | <0.001 | 244 (11) |
| IFNB1 | cytokine | Inhibited | -3.806 | <0.001 | 286 (13) |
| IRF5 | transcription regulator | Inhibited | -3.763 | <0.001 | 357 (17) |
| IFNA1/IFNA13 | cytokine | Inhibited | -3.538 | <0.001 | 103 (10) |
| TGM2 | enzyme | Inhibited | -3.516 | <0.001 | |
| SREBF2 | transcription regulator | Inhibited | -3.482 | <0.001 | 250 (5) |
| TRIM24 | transcription regulator | Activated | 4.884 | <0.001 | 68 (6) |
| MAPK1 | kinase | Activated | 4.26 | <0.001 | 106 (8) |
| PNPT1 | enzyme | Activated | 4.123 | <0.001 | 260 (7) |
| NKX2-3 | transcription regulator | Activated | 3.788 | <0.001 | |
| INSIG1 | other | Activated | 3.628 | <0.001 | 190 (5) |
| ACKR2 | G-protein coupled receptor | Activated | 3.606 | <0.001 | 233 (5) |
| PTGER4 | G-protein coupled receptor | Activated | 3.286 | <0.001 | 363 (10) |
| MYC | transcription regulator | Activated | 3.202 | <0.001 | 231 (2) |
| SIRT1 | transcription regulator | Activated | 3.184 | 0.0582 | |
| SOCS1 | other | Activated | 3.18 | 0.015 | |
| MFSD2A | transporter | Activated | 3.148 | <0.001 | |
| IL1RN | cytokine | Activated | 3.048 | <0.001 | 362 (11) |
| LEPR | transmembrane receptor | Activated | 3 | 0.219 | |
| Mek | group | Activated | 2.992 | 0.475 | |
| MMP3 | peptidase | Activated | 2.828 | 0.351 | |
| IRF8 | transcription regulator | Activated | 2.819 | 0.016 | |
| JQ1 | chemical reagent | Activated | 2.779 | 0.055 | |
| USP18 | peptidase | Activated | 2.772 | <0.001 | 72 (9) |
| POR | enzyme | Activated | 2.729 | <0.001 | |
| IKZF3 | transcription regulator | Activated | 2.72 | 0.005 | |
| HMGA1 | transcription regulator | Activated | 2.718 | 0.337 | |

Signalling", "Death Receptor Signalling" and "Induction of Apoptosis by HIV1" showed negative Z scores of -2.8, -2.3 and -1, respectively (S3 File): predicting lower capacity for apoptosis/cell death in Holstein infected cells. In contrast, "FAT10 (UBD) Cancer Signalling Pathway" (Z score +2.6), "Small Cell Lung Carcinoma" (Z score +1.89), "Non-Small Cell Lung Cancer Signaling" (Z score +2.36), "PI3K Signalling in B lymphocytes" (Z score + 0.8) and "Wnt/Ca

+ Pathway" (Z score +0.7) predicted elevated pathway activation for Holstein infected cells. "FAT10 cancer signalling" with the higher expression of *UBD* (*FAT10*) predicted to promote carcinogenesis via NF-κB and TGFB is illustrated (S1 Fig). In addition, four of the top fifteen entries with a negative Z score in the disease processes or molecular function analysis (Table 2) were placed in "Cell Death and Survival", while the majority (twelve) of the top twenty activated (positive Z score) disease processes/functions were in "Cancer" or "Cellular movement".

Canonical pathways associated with oncogenesis displaying a positive Z score were not identified in analysis of the infection associated-H/S data set. However, infection associated genes known to function in oncogenesis were identified as displaying higher expression in Holstein infected cells compared to Sahiwal. These genes include *UBD* (*FAT10*) and *SLIT2* (see Table 1 and S1 Fig). *PRUNE2* (B-cell CLL/lymphoma 2 and adenovirus E1B 19 kDa interacting family) and *CLU*, encoding Clusterin a secreted chaperone, were also identified and have been shown to operate in cell death, cell transformation, tumour progression and neurodegenerative disorders [57–59].

Taken together the results of the IPA analysis indicate for Holstein infected cells relative to Sahiwal a higher oncogenic potential, but lower innate immunity and cholesterol biosynthesis potential. And for all three of these processes, a clear link to genes whose expression is altered by parasite infection of leukocytes was established

## Differential expression of genes encoding predicted secreted/receptor proteins

In our dataset of H/S-DE genes, we found 83 predicted to encode secreted/extracellular proteins, with 28 of these identified as infection associated (S5 File). The most significantly elevated of these genes was *SLIT2*. Genes encoding CLU, SPARC and PLAU, were also higher more than 1fold, $\log_2$ in Holstein cells and modulated by parasite infection. SPARC (cysteine rich acidic matrix associated protein) promotes both metastasis [60] and degradation of p53 [61], which also occur in *Theileria* infected leukocytes [13,62]. *PLAU* encodes a secreted urokinase associated with cancer progression/metastasis via upregulation by the AT hook factor, HMGA1 [63]. Infection associated genes encoding secreted proteins that showed lower relative expression > 1fold, $\log_2$ in Holstein cells were *ADMATS18*, *C1QTNF6* (complement C1q tumour necrosis related protein: anti-inflammatory), *SPOCK2* (ERK signalling pathway, procancer), *TNFSF10* (cytokine that induces apoptosis in transformed cells) and *CCL5* (chemokine involved in inflammation and immuno-regulation; chemo-attractant for blood monocytes and memory T helper cells). A number of non-infection associated genes that encode secreted proteins with potential to influence the phenotype of the infected cell and/or cells of the host immune system were also identified (e.g. *IL9*, *CXCL2*, *CLEC3B* and *ADAMTS19*). Therefore, differential stimulation of and interaction with cells of the haematopoietic system by factors secreted by Holstein relative to Sahiwal infected cells is likely.

Numerous genes encoding cellular receptors were also present in the H/S DE data set (see S6 File). Infection associated receptor encoding genes include *CCR1* (encoding a receptor for CCL5), *CRLF2* (cytokine receptor like factor *2*), *IL10RA*, *TLR2*, *CD2*, *CD48* and *IL7R*. Non-infection associated genes encoding receptors included *ROBO2*; encoding the receptor for the secreted SLIT2 factor (both receptor and factor expressed higher in infected Holstein cells), *TLR7*, *CD4*, *TRAF1* (TNF receptor associated factor 1), *IL27RA*, *IL17RD* and *IRAK1* (Interleukin-1 receptor-associated kinase 1). Therefore, Holstein infected cells are predicted to possess the potential to react differently than Sahiwal, in response to a large array of receptor ligands, including chemokines, cytokines and growth factors.

## qRT-PCR validation of differential expression of candidate genes

To confirm the validity of the RNA-seq data, qRT-PCR analysis was performed. Seventeen genes were selected based on their representative RNA-seq profiles across the H/S-DE data set. The results are summarized in Table 4 and shown for a subset of the genes in Fig 5. *SLIT2* and *PRUNE2* represent infection-associated genes where expression was assessed as higher by RNA-seq in Holstein relative to Sahiwal infected cells. qRT-PCR confirmed this profile (Table 4), with statistically significant higher mRNA levels for both genes, with on average 62.1- and 171.6-fold higher values for *SLIT2* and *PRUNE2* from Holstein infected cell lines, respectively. *IFI44*, *ISG15*, *MX1*, *RSAD2*, *SAMD9* and *TNFSF10* were chosen as they are infection-associated genes that show lower expression levels in Holstein infected cells compared to Sahiwal cells. In all six cases the qRT-PCR supported this profile with a statistically significant lower mean expression level in Holstein cells of 8.9-fold (*IFI44*), 22.2-fold (*ISG15*), 17.5-fold (*MX1*), 6.1-fold (*RSAD2*), 5.9-fold (*SAMD9*) and 3.4-fold (*TNFSF10*), respectively (Table 4, Fig 5A & 5B). *ADAMTS20*, *NECTIN3* and *WLS* were selected to represent non-infection associated genes with higher expression in Holstein cells. The qRT-PCR analysis confirmed this profile with statistically significant higher relative expression of 2.0 fold (*ADAMTS20*), 9.0-fold (*NECTIN3*) and 5.4-fold (*WLS*) (Table 4, Fig 5C & 5D) in Holstein cells compared to Sahiwal cells. The reciprocal profile was also validated with *HERC5*, *IFI6*, *CGAS (MB21D1)* and *OAS1Y*. Statistically significant lower expression of 3.3-fold (*HERC5*), 6.0-fold (*IFI6*), 2.2-fold (*CGAS*) and 10.5-fold (*OAS1Y*) was found in Holstein infected cells compared to Sahiwal cells (Table 4). Thus, for the majority of genes in the RNA-seq data set, the trend of expression predicted is likely to be accurate. Two further genes encoding the pro-inflammatory cytokines, IL6 and IL23A, designated as significantly higher in the H/S DE data set but which showed non-consistent differential expression across the eleven samples, were tested. The results indicated that although the mean expression values were 1.6-fold *(IL6)* and 3.6-fold *(IL23A)* higher

**Table 4. qRT-PCR validation of differential expression of a subset of H/S-DE genes.**

| Gene | Gene Symbol | Sahiwal | Holstein | P value |
|---|---|---|---|---|
| ADAM Metallopeptidase With Thrombospondin Type 1 Motif 20 | ADAMTS20 | 1.3 ± 0.1 | 2.6 ± 0.4 | 0.011 |
| Cyclic GMP-AMP Synthase | CGAS | 2.8 ± 0.2 | 1.2 ± 0.1 | <0.001 |
| HECT And RLD Domain Containing E3 Ubiquitin Protein Ligase 5 | HERC5 | 3.7 ± 0.6 | 1.1 ± 0.1 | <0.001 |
| Interferon Alpha | IFNA | 1.7 ± 0.3 | 2.1 ± 0.4 | n.s. |
| Interferon Alpha Inducible Protein 6 | IFI6 | 11.8 ± 1.1 | 2.0 ± 0.4 | 0.001 |
| Interferon Beta 1 | IFNB1 | 2.3 ± 0.4 | 5.7 ± 1.1 | 0.006 |
| Interferon Beta 3 | IFNB3 | 5.1 ± 1.4 | 2.9 ± 0.7 | n.s. |
| Interferon Induced Protein 44 | IFI44 | 50.0 ± 8.4 | 5.6 ± 1.4 | 0.002 |
| Interleukin 6 | IL6 | 2.8 ± 0.6 | 4.5 ± 1.8 | n.s. |
| Interleukin 23 Subunit Alpha | IL23A | 1.6 ± 0.2 | 5.8 ± 3.8 | n.s. |
| ISG15 Ubiquitin Like Modifier | ISG15 | 110.8 ± 26.3 | 5.0 ± 1.8 | 0.001 |
| MX Dynamin Like GTPase 1 | MX1 | 191.2 ± 14.4 | 10.9 ± 8.1 | 0.005 |
| Nectin Cell Adhesion Molecule 3 | NECTIN3 | 5.4 ± 2.0 | 48.3 ± 13.8 | 0.018 |
| 2'-5'-Oligoadenylate Synthetase 1 | OAS1 | 231.0 ± 23.5 | 22.0 ± 17.9 | 0.001 |
| Prune Homolog 2 With BCH Domain | PRUNE2 | 2.0 ± 0.6 | 335.3 ± 172.9 | 0.018 |
| Radical S-Adenosyl Methionine Domain Containing 2 | RSAD2 | 153.4 ± 35.1 | 25.1 ± 19.2 | 0.013 |
| Sterile Alpha Motif Domain Containing 9 | SAMD9 | 8.5 ± 1.5 | 1.4 ± 0.1 | 0.001 |
| Slit Guidance Ligand 2 | SLIT2 | 2.1 ± 0.5 | 127.4 ± 65.0 | 0.007 |
| TNF Superfamily Member 10 | TNFSF10 | 12.5 ± 1.6 | 3.7 ± 0.9 | 0.009 |
| Wnt Ligand Secretion Mediator | WLS | 1.2 ± 0.1 | 6.7 ± 1.7 | 0.008 |

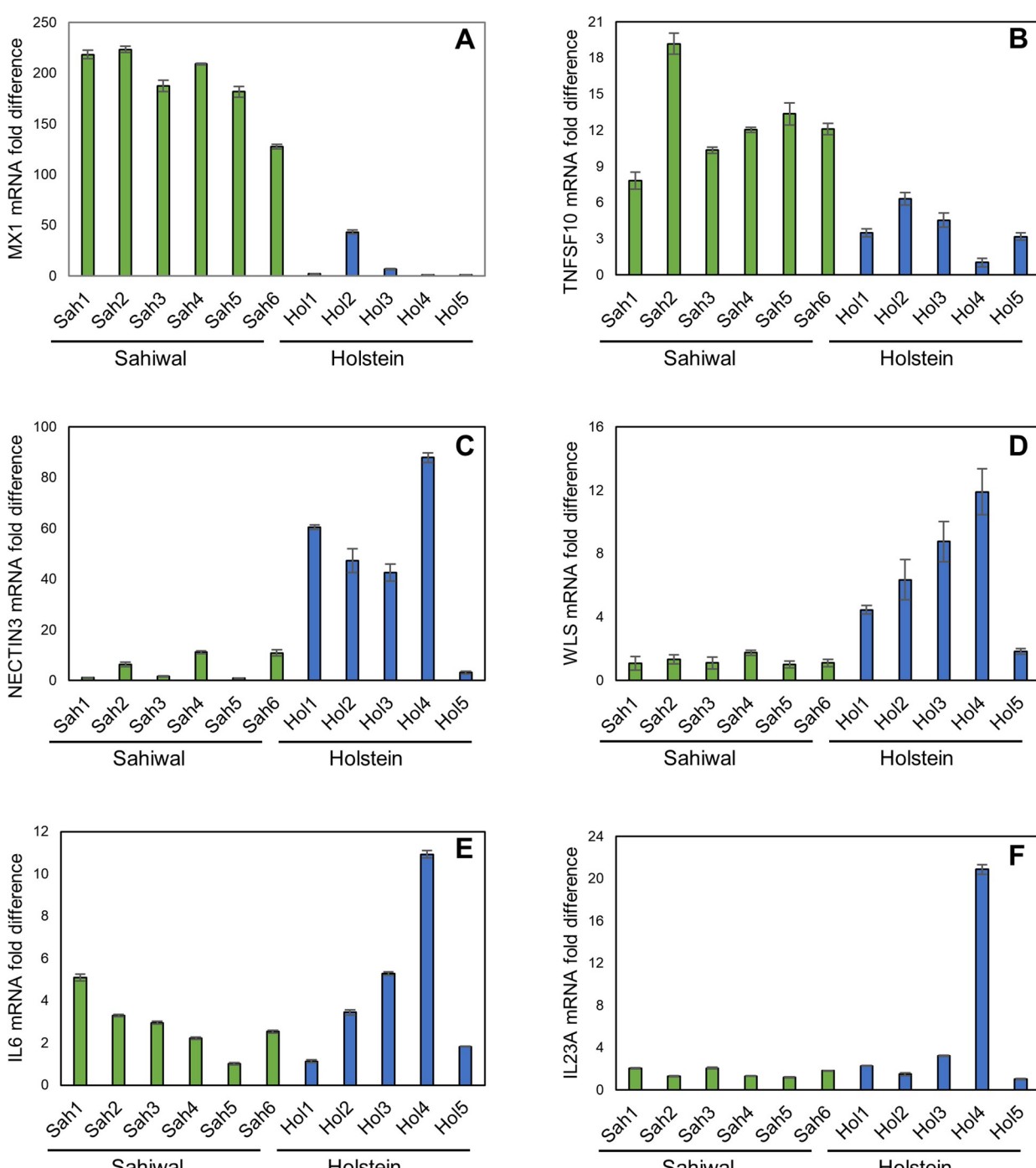

**Fig 5. qRT-PCR validation of candidate H/S-DE genes.** Panels show results for representative genes. (A) *MX1*; (B) *TNSF10*; (C) *NECTIN3*; (D) *WLS*; (E) *IL6*; (F) *IL23A*. Within each panel Y axis shows mRNA fold difference (calculated *vs.* the lowest expressed sample); X axis designates sample number and breed; Hol1-5, Holstein; Sah1-6, Sahiwal.

for Holstein infected cells relative to Sahiwal, variability within the data precluded validation of a significant difference (Table 4, Fig 5E & 5F).

IPA analysis of our RNA-seq datasets predicted marked differences between Holstein and Sahiwal infected cells in pathways linked to activation of and response to type I IFN. However,

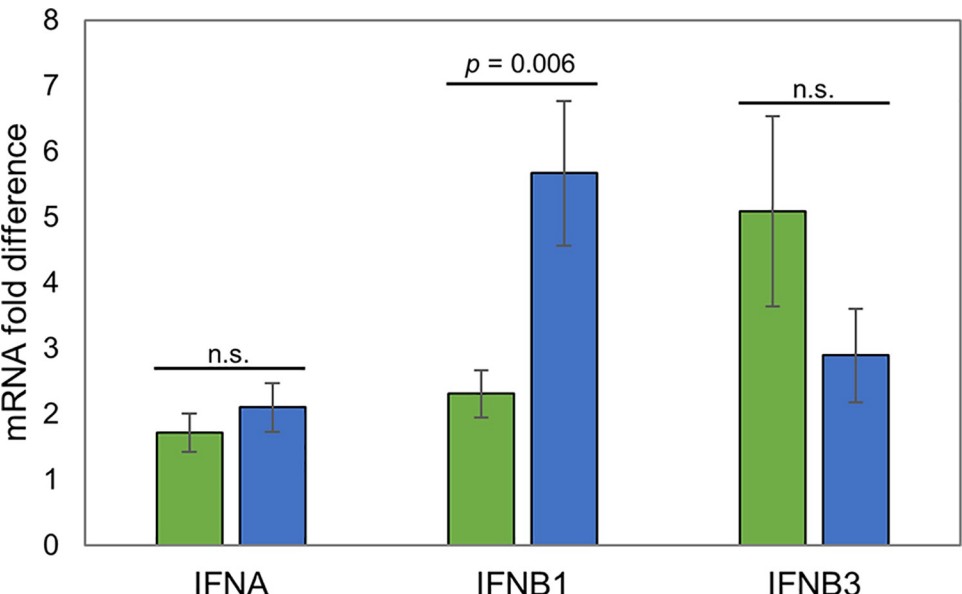

**Fig 6. qRT-PCR analysis of INF genes: *INFA*, *INFB1* and *INFB3*.** The Y-axis shows mRNA fold difference (calculated *vs.* the lowest expressed sample): Green infill designates, Sahiwal; blue infill, Holstein. Degree of significance denoted above bar.

no type I IFN genes were identified as differentially expressed in the H/S-DE dataset. In an attempt to elucidate whether a type I IFN could be associated with differential expression of ISGs, the expression of *IFNB1*, *IFNB3* and *IFNA* was investigated by qRT-PCR. No significant difference in *IFNA* and *IFNB3* mRNA levels were detected between Holstein and Sahiwal derived infected cells. However, there was a significant difference in *IFNB1* expression. Unexpectedly, on average 2.5-fold higher expression of *IFNB1* was detected in Holstein infected cells compared to Sahiwal cells (Table 4, Fig 6).

## Screen for differential expression of parasite transformation candidates and motifs bound by TashAT2 in *B. indicus vs. B. taurus* genomes

In total, 109 *T. annulata* encoded genes were identified as significantly differentially expressed between Holstein and Sahiwal infected cell lines (S7 File). Compared to host genes, however, the parasite genes showed markedly smaller magnitudes of change in expression level; a maximum $log_2$ fold change of 2.6 compared to -26.1 for the bovine gene set. A large number of the parasite genes were annotated as hypothetical or encode proteins with a predicted function/location that cannot be linked to transformation of the infected cell or host immune response. Moreover, there was limited evidence for altered expression of genes encoding candidate modulators of host cell phenotype (TaPIN, Ta9 and TashATs). Thus, gene *TA15705*, encoding the Ta9 immuno-dominant antigen [17], was the only one identified as differentially expressed (0.84 $log_2$ fold higher in Holstein, third most significant). Further analysis of Ta9 expression was performed by immunoblot. The result failed to indicate significant elevation in Ta9 protein levels across three of the Holstein lines relative to three Sahiwal (S2 Fig). A similar result was obtained for TashAT2 translocated to the host nucleus of the infected cell [44]. Immunofluorescence failed to detect a consistent difference in reactivity against the infected cell nucleus between all three Holstein lines relative to Sahiwal (S3 Fig).

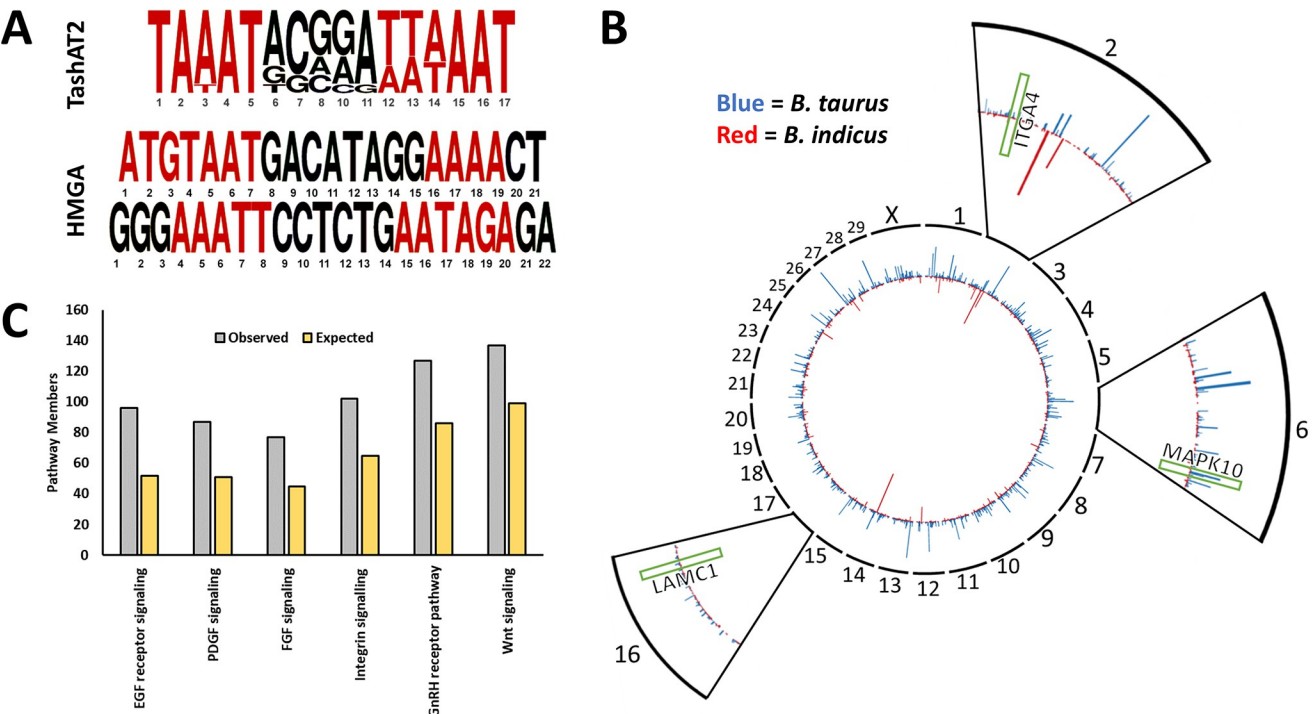

**Fig 7. Analysis of DNA motifs bound by the parasite encoded TashAT2 factor in the genomes of *B. indicus vs. B. taurus*.** (A) Consensus motif of the DNA region bound by parasite TashAT2 in the genome of *B. indicus vs. B. taurus* compared to the binding motifs of mammalian HGMA [19]. AT rich regions bound by AT hooks found in both TashAT2 and HGMA are shown in red. (B) A CIRCOS summary plot with automatic binning showing the number of TashAT2 binding motifs found in genes across the *B. indicus* (red) and *B. taurus* (blue) genomes. Genes with the same number of motifs are excluded to demonstrate the pattern of differences between each breed. Zoomed views are shown for chromosomes 2, 8 and 16 without automatic binning. The approximate regions of genes in the "EGF" and "Integrin signalling" pathways that possess different numbers of TashAT2 binding motifs and differ in expression between infected Holstein and Sahiwal breeds are indicated in green (*ITGA4*, *MAPK10* and *LAMC1*). (C) Overrepresented PANTHER pathways in the cohort of bovine genes with FDR adjusted p value < 0.05 after a Fisher's Exact test. Blue is the observed members of a pathway in the dataset, whereas red indicates the expected number if the distribution was random.

Polymorphism of a secreted parasite factor, or diversity of the host target it interacts with, could differentially modulate host cell gene expression in the absence of differential expression, and this model has been predicted for TashAT proteins. Thus, for TashAT2/3, polymorphism has been identified in the number and pattern of their AT hook DNA binding domains [64]. The AT hook domain of TashAT2/3 shows similarity to that of mammalian HMGA proteins, suggesting potential overlap in target DNA motifs, and it is known that the TashAT2 domain recognises motifs rich in AT (Fig 7A) [19]. Our IPA upstream analysis predicted activation of HMGA1 (Table 3), and HMGA1 has been shown to lower expression of cholesterol biosynthesis genes [55], one of the most enriched pathways in our data sets. Therefore, we assessed whether the pattern of motifs bound by TashAT2 show diversity in the genomes of *B. indicus* relative to *B. taurus* and whether they might be enriched in our IA-H/S data set. Our initial screen identified 167,151 incidences of the target motifs in the *B. taurus* and 149,107 in the *B. indicus* genome (S8 File). The majority of the motif patterns in both genomes were located outside of gene coding regions (≈ 56% and ≈65% in *B. taurus* and *B. indicus* respectively), while ≈42% were within introns in *B. taurus* and ≈35% in *B. indicus*. Approximately 2% of patterns were in upstream or downstream UTRs and less than 1% were within exons in each breed (S8 File). As motif sites located within gene coding regions are more likely to be cis-acting elements and their associated targets more easily inferred, we used the cohort of motifs located in

UTRs, introns and exons for a more detailed comparison (S9 File). In *B. taurus* 11,983 genes featured at least one TashAT2 binding motif compared to 9,516 in *B. indicus* and there were substantial differences in the pattern and number of TashAT2 binding motifs between breeds (Fig 7B; S9 File). There were 7,575 shared genes that possessed at least one TashAT2 binding motif and it was evident that for certain genes (including *SLIT2*, *PRUNE2*, *LITAF* and *NEC-TIN3*) the number of motifs differed between *B. indicus* and *B. taurus* genomes. Subsequent PANTHER analysis of the shared genes data set showed an enrichment in genes involved in "EGF (PI3K/AKT, ERK) receptor signalling", "PGDF signalling", "Wnt signalling pathway", "FGF signalling", "Gonadotropin-releasing hormone (GnRH)" and "Integrin signalling" (FDR adjusted $p < 0.05$; Fig 7C). Comparison with IPA on the H/S-DE gene set showed overlap with "ERK/MAPK signalling", "PI3K/AKT signalling", "Wnt/Ca+ pathway and "Integrin signalling" (S3 File). Additionally, of the 517 transcripts previously noted to be altered by parasite infection of BL20 cells and differ in expression between infected Holstein and Sahiwal cells, 232 (45%) feature TashAT2 binding motifs. This is significantly more than would be expected by chance under a binomial distribution in which $\approx$26% of annotated features in the bovine genome possess a binding motif ($p < 0.001$). *SLIT2*, *LITAF*, *PRUNE2* and genes involved in "Integrin signalling" showed altered numbers of TashAT2 motifs (S10 File) and, together with "PI3K/AKT signalling", integrin signalling genes were significantly enriched by IPA in the Infection Associated-H/S data set (S4 File).

## Discussion

Understanding how differential susceptibility to infectious disease is conferred is of great importance. Such understanding is needed to highlight at risk groups, inform therapeutic regimes and develop strategies for breeding productive but infection tolerant livestock. To gain insight on how variable infection responses are generated, we have utilised two cattle breeds representing the ends of a susceptibility spectrum to tropical theileriosis, caused by *Theileria annulata*. The Sahiwal breed (*B. indicus*) has evolved in tandem with the pathogen and in general, displays mild symptoms of disease when infected, whereas Holstein (*B. taurus*) are generally susceptible to acute disease. Based on the premise that tolerance has most likely evolved via competitive molecular interactions between pathogen and host that lead to pathology, we focused on identifying gene expression differences between low passage *T. annulata* macroschizont infected cell lines from tolerant Sahiwals and susceptible Holsteins, and screened for parasite factors that could generate these differences.

RNA-seq analysis identified a large number of genes displaying a significant difference in expression level between Holstein and Sahiwal infected cells (the H/S-DE data set). In order to replicate, as closely as possible, the situation *in vivo* and to prevent loss of virulence through cloning, this data set was generated from uncloned, low passage infected cell lines derived from 11 different animals. PCA analysis of the data set showed that although the two breed types clearly separated there was an inherent level of variability for samples within a breed. The result was not unexpected. It has been shown that gene expression levels, including cytokine encoding genes, vary within breed for both Holstein and Sahiwal infected cell lines [13,21,42], and gene expression markers associated with virulence show appreciable variability [65]. Such within breed variability could be generated by: a) genetic differences between individual cattle, with variance likely to be higher for the Holstein (genetically unrelated) relative to the Sahiwal cattle (full and 1/2 siblings) [9]; b) genetic differences in parasite genotype represented by infected cell lines, because although all cell lines are infected by Hissar, this strain contains multiple parasite genotypes [65]; c) epigenetic differences generated during establishment of cell lines *in vivo* or *in vitro* and d) variable host cell type composition between infected

cell lines. We propose that the gene expression profile of infected cells may be unique for each individual animal and that this influences the wide variability in infection outcome between and within "tolerant" and "susceptible" breeds [9,66]. To validate this premise, analysis of multiple pairs of infected and uninfected cells from a range of "tolerant" and "susceptible" breeds is required.

Despite the expectation of variable expression profiles, a large number of genes in our H/S-DE data set displayed expression differences that were consistent across all Sahiwal *vs*. Holstein samples (see denoted genes, S2 File); while the remainder, as indicated above, showed a higher level of variability between individuals across the breeds (for example see *IL-6*). The results support the conclusion that there is a major influence of breed type on the phenotype of the macroschizont-infected leukocyte linked to disease susceptibility [2,5,13].

IPA was performed on both the full H/S-DE data set and the more limited set (IA-H/S) of genes whose expression is induced by parasite infection of the host leukocyte. Strikingly the most significantly enriched pathways for both data sets were the same. However, previous studies indicate that a number of infection-associated genes differentially expressed between breeds have been missed by our analysis, and there are at least several candidates. Thus, the *TGFB2* gene, that is induced by infection and expressed at a higher level in Holstein infected cells [13], was present in our H/S-DE data set (elevated 3.5 log$_2$ fold) but was not identified as infection associated. *ICAM1* was reported as differentially expressed between Sahiwal and Holstein infected cells [23] but was not highlighted as significantly different in our study. The reasons for these false negatives are not conclusively known, but most likely relate to differences in bovine cells used to establish changes linked to infection and methodology for identification of differential gene expression.

The pathway showing the most consistent and significant modulation across our data sets was type 1 IFN. Modulation was most evident as lower relative expression of ISG in Holstein infected cells (82% of identified ISGs) but was also recorded for genes encoding pattern recognition receptors (DDX58 [RIG-1], ZBP1 [DAI], IFIH1 [MDA-5] and CGAS) and transcription factors (STAT1, STAT2, IRF3, IRF7 and IRF9). Notably, for many ISG genes differential expression was consistent across all Holstein *vs*. Sahiwal samples. Validation of gene expression differences was obtained by qRT-PCR for several ISG genes (*MX1*, *OAS1Y*, *ISG15*, *RSAD2* [Viperin], *IFI6*, *IFI44* and *HERC5*), and a number of ISG genes (*HERC6*, *ISG15*, *IFI6*, *IFI44*, *IFIH1*, *IFI35*, *MX1*, *MX2*, *OAS2* and *ZBP1*) were identified as modulated by parasite infection, supporting the previous report on ISG15 [20]. The function of ISG genes in innate immunity has been studied predominantly using viral systems (reviewed in [67,68]). Limited information regarding the function of ISGs against intracellular protozoa is available. Nevertheless, it is reasonable to propose that, by acting as tumour suppressors or regulators of immune effector cells, ISGs could modulate the phenotype of the *Theileria*-infected leukocyte and influence infection outcome [69–74]. Modulation of ISG expression between cattle breeds could also be of relevance to other pathogens. An ISG response is generated in infected cells against Foot and Mouth Disease Virus [75] and *B. indicus* cattle are known to be less susceptible to FMDV infection than *B. taurus* [76].

While IPA predicted modulation of type I IFN production associated with tolerance, there was no direct evidence for this in the RNA-seq data set. This may have been due to difficulty in mapping RNA-seq reads to specific IFN genes within the large type I IFN family [77]. However, contrary to the IPA network prediction, qRT-PCR indicated significantly higher levels of *IFNB1* expression in the Holstein cell lines, supporting previous data demonstrating expression of *IFNB* in infected leukocytes [78]. How differential expression of *IFNB1* is generated between the infected cell lines is currently unclear but could occur via lower expression of genes involved in cholesterol biosynthesis recorded for Holstein infected cells (see Fig 5). A

reduction in cholesterol biosynthesis has been linked to induction of *IFNB* expression and inflammatory disease [52,79] and is implicated in the induction of an inflammatory cytokine storm in COVID-19 [80]. It appears, therefore, that tolerance to tropical theileriosis is associated with a higher relative ISG expression in Sahiwal infected cells; while disease susceptibility of Holstein may be linked to elevated levels of IFNB1 cytokine production and a stronger propensity to stimulate an inflammatory response. Whether these events are associated with the massive pulmonary oedema following metastasis of infected cells to the lungs [12] requires investigation. Parallels with mechanisms conferring susceptibility to other infectious diseases are of interest.

IPA also identified several processes linked to oncogenesis. This included a prediction of greater metastasis potential for Holstein infected cells (via, for example elevated *FAT10* [*UBD*], *TGFB2*, *ITGA1*, *PLAU* and *SPARC* gene expression), supporting and validating the previous work demonstrating a role for TGFB2 in stimulating elevated metastasis of Holstein relative to Sahiwal infected cell lines [13]. Previous KEGG pathway analysis of microarray data led to postulation that disease susceptibility is linked to an aberrant interaction with cells of the immune response that promotes immune dysfunction [23]. Major differences in expression of genes associated with immune cell interaction (such as receptors, chemokines and cytokines), highlighted in our current analysis, supports this premise. Thus, reduced oncogenic potential and lower aberrant immune cell activation could account for the less pronounced lymph node enlargement observed in infected Sahiwal relative to Holstein cattle [9].

A major aim of this study was to investigate potential pathogen-host interactions that generate gene expression differences between infected cells from tolerant *vs*. susceptible breeds. One potential interaction previously considered is that the type of cell preferentially infected could differ between *B. indicus* and *B. taurus* breeds. Analysis with antibodies against cell type markers indicated that myeloid cells were the predominant infected cell type infected for both breeds [81], although evidence was presented for some macroschizont infected NK cells in Sahiwal lines. Our RNA-seq data support these findings. For genes encoding the myeloid lineage markers, CCR2, CCR3, CD13, CD11, CD14, CD34, CD36, CD9 and CD38, none were identified as differentially expressed. For genes encoding the CD8 and CD2 markers previously identified on some Sahiwal infected cells [81], *CD8* was not identified as significant in our data set. *CD2* showed higher expression levels associated with Sahiwal line, although in three Holstein lines the level of expression was close to that of one of the Sahiwal lines. In contrast, the gene encoding the CD4 marker found on T cells and myeloid cells showed significantly lower expression values in all Holstein cell lines (-4.6 log$_2$). The results imply that either the profile of cell types infected can vary to a degree between *B. indicus* and *B. taurus*, or following transformation of predominantly myeloid cells, the expression of cell surface molecules is differentially modulated. Previous analysis of infected, purified myeloid cells [23] also highlighted lower expression of *CD4* in Holstein relative to Sahiwal cells, and together with alteration of surface marker expression following infection of the BL20 line [14,15], indicates the latter possibility as most likely.

Interactions that modulate gene expression between Sahiwal and Holstein infected cells are likely to involve pathogen modulators of host cell gene expression. No conclusive evidence of this for known parasite candidates was detected. While the gene encoding Ta9 reported to activate AP1 [17] was scored as significantly expressed at a higher level (0.84 log$_2$ fold) in Holstein infected leukocytes, this was not validated at the protein level by immunoblot. IPA, however, did identify HMGA1 as a potential regulator of differential gene expression between the two breeds. Mammalian HMGAs act as architectural transcription factors and bind patterns of AT rich motifs to alter chromatin structure (both locally and globally) [82]. HMGA factors regulate gene expression primarily during development [83], but also in neoplasia [84,85] and

genes targeted by NF-κB [86]. Given that TashAT2 is known to encode a DNA binding domain with homology to HMGAs, binds a related AT rich motif pattern (Fig 7A) and was detected in the host nucleus of both Sahiwal and Holstein infected cells (S3 Fig), we screened the available *B. indicus* and *B. taurus* genomes for the presence of motifs bound by TashAT2. The results clearly demonstrated a difference in the predicted pattern of these motifs (Fig 7B). Thus, genes in both genomes uniquely display a predicted TashAT2 bound motif, while for genes that shared possession of motifs, differences in motif number were frequently detected. Within this cohort of genes there are many that could influence the infected cell phenotype. For example, there is enrichment for genes that function in "Epidermal Growth Factor signalling" and overlap with pathways highlighted by IPA of the H/S-DE data set ("ERK", "PI3K/ AKT", "Integrin signalling"). PI3K/AKT signalling has previously been linked to mechanisms involved in transformation of the *Theileria*-infected leukocyte [55,87] and is modulated by mammalian HMGA2 proteins [84]. Integrin signalling is well established in oncogenesis and cellular interaction [88,89], and integrin genes are targets for modulation by HMGA [85]. It was also demonstrated that 45% of infection-associated genes differentially expressed between Sahiwal and Holstein infected cell lines feature at least one putative TashAT2 binding motif, which is more than expected by chance (p < 0.001). "PI3K/AKT" and "Integrin signalling" were significant in IPA of the IA-H/S data set (S4 File) and several infection-associated genes involved in integrin signalling, possess differences in the number of TashAT2 binding motifs between the two genomes (LAMC1, ACTA2, ITGA4 and ITGB5). The majority of integrin genes are expressed at a lower level in Sahiwal infected cells relative to Holstein. Reduced expression of ITGA4 via PI3K signalling has been linked previously to a loss of infected cell virulence [87].

Based on the above results, and on the premise that AT hook DNA binding proteins act as base composition readers to mould the structure of the epigenome [90], we propose the following model. Differences in the pattern of AT rich DNA motifs between the genomes of tolerant *vs*. susceptible cattle breeds (and individuals), when bound by pathogen TashAT factors located in the host cell nucleus, give rise to alternative chromatin architectures. These architectures then allow variable accessibility of host transcription factors activated by infection, resulting in differential modulation of gene expression in the macroschizont infected cell. Given the known association between the infected cell and pathology, and the polymorphism identified in the TashAT2/3 DNA binding domain [64], this model could explain the considerable variability in disease severity between and within breeds that occur upon infection [9,66]. Validation of differential TashAT2 factor binding to sites in host DNA by chromatin immunoprecipitation, and further investigation of divergent pathogen-host interactions that have evolved to modulate host cell gene expression will provide understanding of how differential susceptibility to disease has arisen, and inform strategies aiming to breed productive livestock tolerant to infection.

## Supporting information

**S1 Fig. IPA canonical pathway, "FAT10 cancer signalling".** The nodes represent genes/molecules/complexes in a pathway, and the lines and arrows between nodes indicate known relationships from the Ingenuity Knowledge Base. Nodes with purple outline indicate molecules that were measured as differentially expressed in our dataset, with the intensity of coloured infill indicating the level of up–(red) or down-(green) regulation of Holstein relative to Sahiwal. The blue and orange coloured molecules and lines are predicted activation states generated by the Molecular Activity Predictor function in Ingenuity Pathway Analysis. Blue colour indicates a predicted inhibition, and orange a predicted activation state in Holstein relative to

Sahiwal. Broad lines with explanatory text beside the pathway indicate the cellular location of molecules in the pathway. The molecules in the pathway are given shapes that indicate their functional class (Nested Circle/Square = Group/Complex, Horizontal ellipse = Transcriptional Regulator, Vertical Ellipse = transmembrane receptor, Vertical Rhombus = enzyme, Square = Cytokine/Growth Factor, Triangle = Kinase, Vertical Ellipse = Transmembrane Receptor, Circle = other). The edges between molecules are also differentiated to indicate the type of relationship between them. Solid lines are direct relationships and dashed lines are indirect.
(TIF)

**S2 Fig. Immunoblot carried out with protein extracts prepared from 3 independent Sahiwal lines compared to 3 Holstein lines.** Sahiwal samples are denoted SA, SB, SC; Holstein denoted HA, HB, HC. Protein size markers are indicated on the right (kDa). **A**. Extracts probed with Rat anti-Ta9 (*TA15705*) at 1/1200 dilution. Ta9 was detected at a variable size of 43-46kDa, which is a similar to that described previously for the polymorphic Ta9 antigen [17,43]. **B**. As a control, the same blot was reprobed with Rabbit antiserum raised against constitutively expressed ER HSP90 (*TA06470*) at 1/1500 dilution. The Ta9 and HSP90 reactive proteins are denoted by arrow.
(TIF)

**S3 Fig. Immunofluorescence assay carried out on cells prepared from 3 independent Sahiwal lines compared to 3 Holstein lines.** Sahiwal samples are denoted SA, SB, SC; Holstein denoted HA, HB, HC. Cells were reacted with antiserum specific for TashAT2 (EL24) and images obtained using matched exposures. Bar = 7 μm.
(TIFF)

**S1 Table. Details of oligonucleotide primers used in qRT-PCR.** F and R denote forward and reverse primers respectively. *Primers designed based on Ensembl transcript.
(DOCX)

**S2 Table. Number of RNA-seq reads mapped on to *B. taurus* genome are not statistically different.** Rows denote RNA-seq sample set derived from 6 Sahiwal (S1-6) or 5 Holstein (H1-5) infected cell lines. Columns denote the % of reads mapped to the *B. taurus* genome. Summary denotes mean % of reads mapped, standard deviation and no significant difference between means.
(DOCX)

**S3 Table. Top upstream regulators with predicted activated or inhibited activation states in infection-associated H/S-DE genes.** In total 655 upstream regulators were found with a p-value of < 0.05; including 123 with a significantly predicted activation state (activated or inhibited, based on a z-score >2, or <-2). Only the top 20 activated and top 20 inhibited regulators based on z-score are presented in the table.
(DOCX)

**S1 File. Excel spread sheet of 2211 H/S-DE genes (<0.1 FDR).** Columns designate: Ensemble Gene ID; base mean of RNA-seq counts across all samples; $\log_2$ fold change between mean of Holstein *vs*. Sahiwal sample counts; lfcSE (Standard Error of the log fold change); stat (Wald statistic); p value (unadjusted p value for the Wald test); padj (Benjamini-Hochberg adjusted p-value for significance of the Wald test; DESeq2 normalized RNA-seq counts for each of the six Sahiwal samples (Sah 1–6) and five Holstein (Hol 1–6); the gene name; an alternative gene name, when applicable; whether RNA-seq counts were consistent across all Sahiwal samples *vs*. Holstein; Interferon associated genes, as identified in the study of Liu et al. [50] or by GO

[37]; description of protein/factor encoded by the gene. Yellow highlight designate genes also modulated in *Theileria* infected (TBL20) cells.
(XLSX)

**S2 File. Excel spread sheet of the 517 genes in H/S-DE that overlap with genes designated as infection associated (modulated in TBL20 *vs.* BL20).** Columns designate: Gene id; DESeq normalized base mean of counts in A (Sahiwal) samples; base mean counts in B (Holstein) samples; log$_2$ fold change between mean of Holstein *vs.* Sahiwal sample counts; lfcSE (Standard Error of the log fold change); stat (Wald statistic); pvalue (unadjusted p value for the Wald test); padj (Benjamini-Hochberg adjusted p-value for significance of the Wald test; the gene name; description of protein/factor encoded by the gene.
(XLSX)

**S3 File. Excel spread sheet of the 100 most significant canonical pathways designated as enriched by IPA for H/S-DE data set.** Columns designate: The enriched canonical pathway; the–log p value; ratio of the number of genes from the list that maps to the pathway divided by the total number of genes that map to pathway; the (activation) Z-score; the gene symbol for molecules in H/S-DE data set present in the pathway. Blue highlight denotes pathways with a Z score>2, indicating repression in Holstein *vs.* Sahiwal infected cells; red highlight indicates activation (Z-score >1.5).
(XLSX)

**S4 File. Excel spread sheet of the 100 most significant canonical pathways enriched by IPA from the data set of 517 overlapped infection-associated H/S-DE genes.** Columns designate: The enriched canonical pathway; the–log p value; the ratio of the number of genes from the list that maps to the pathway divided by the total number of genes that map to pathway; the (activation) Z-score; the gene symbol for molecules in H/S DE data set present in the pathway. Blue highlight denotes pathways with an activation score (Z score>2) indicating repression in Holstein *vs.* Sahiwal infected cells.
(XLSX)

**S5 File. Excel spread sheet of genes in H/S-DE data set predicted to encode secreted proteins.** Columns designate: Ensemble Gene ID; DESeq normalized base mean of counts in all samples; log$_2$ fold change between mean of Holstein *vs.*Sahiwal sample counts; lfcSE (Standard Error of the log fold change); stat (Wald statistic); pvalue (unadjusted p value for the Wald test); padj (Benjamini-Hochberg adjusted p-value for significance of the Wald test; the gene name; description of protein/factor encoded by the gene; prediction of location based on gene ontology data in Genecards [35]; Yellow highlight designate genes also modulated in *Theileria* infected (TBL20) cells.
(XLSX)

**S6 File. Excel spread sheet of genes in H/S-DE data set predicted to encode receptors.** Columns designate: Ensemble Gene ID; Deseq2 normalized base mean for RNA-seq counts for all Sahiwal and Holstein samples; the log$_2$ fold change between Holstein and Sahiwal samples for that gene; lfcSE (Standard Error of the log fold change); stat (Wald statistic); pvalue (unadjusted p value for the Wald test); padj (Benjamini-Hochberg adjusted p-value for significance of the Wald test; the gene name; description of protein/factor encoded by the gene; Yellow highlight designate genes also modulated in *Theileria* infected (TBL20) cells.
(XLSX)

**S7 File. Excel spread sheet of 109 parasite (*T. annulata*) genes (<0.1 FDR).** Columns designate: TA number (T. annulata gene ID); DESeq normalized base mean of RNA-seq counts

across all samples; designation of whether expression is higher (up) or lower (down) in Holstein (*B. taurus*) samples relative to Sahiwal (*B. indicus*); DESeq normalized base mean of counts in A (Sahiwal) samples; base mean counts in B (Holstein) samples; log$_2$ fold change between mean of Holstein *vs.*Sahiwal sample counts; lfcSE (Standard Error of the log fold change); stat (Wald statistic); pvalue (unadjusted p value for the Wald test); padj (Benjamini-Hochberg adjusted p-value for significance of the Wald test; description of protein/factor predicted to be encoded by the gene; location of gene on *T. annulata* genome; genomic sequence ID.
(XLS)

**S8 File. Excel spread sheet of nucleotide motifs bound by TashAT2 in genome of *B. taurus* and *B. indicus*.** Columns designate chromosome; motif start position; motif end position; genomic feature linked to motif position (non-coding, intron, exon, untranslated region (UTR); gene product associated with motif and genomic feature, if applicable.
(XLSX)

**S9 File. Excel spread sheet of nucleotide motifs bound by TashAT2 in gene coding (RNA) regions of *B. taurus* and *B. indicus* genomes.** The data sheets list motifs identified in *B. taurus* genes, motifs identified in *B. indicus* genes and motifs in genes shared by *B. taurus* and *B. indicus*. Columns indicate: Gene identifier, number of motifs identified in that gene region; and for shared genes: B. number of motifs identified in *B. taurus* gene; C. number of motifs identified in *B. indicus* gene.
(XLSX)

**S10 File. List of genes in integrin signalling pathway that bear different numbers of nucleotide motif bound by TashAT2 in *B. taurus* and *B. indicus* genomes.** Green highlight designates genes that display differential expression between Sahiwal and Holstein infected cells.
(DOCX)

## Acknowledgments

We would like to thank the Centre for Virus Research, University of Glasgow for providing access to the IPA software.

## Author Contributions

**Conceptualization:** Elizabeth J. Glass, Brian R. Shiels.

**Data curation:** Stephen D. Larcombe, Paul Capewell, William Weir.

**Formal analysis:** Paul Capewell, Kirsty Jensen, Jane Kinnaird, Brian R. Shiels.

**Funding acquisition:** William Weir, Elizabeth J. Glass, Brian R. Shiels.

**Investigation:** Stephen D. Larcombe, Kirsty Jensen, Brian R. Shiels.

**Methodology:** Stephen D. Larcombe, Paul Capewell, Kirsty Jensen, William Weir.

**Project administration:** Brian R. Shiels.

**Supervision:** Brian R. Shiels.

**Validation:** Stephen D. Larcombe, Kirsty Jensen, Jane Kinnaird.

**Writing – original draft:** Stephen D. Larcombe, Paul Capewell, Brian R. Shiels.

**Writing – review & editing:** Kirsty Jensen, William Weir, Jane Kinnaird, Elizabeth J. Glass, Brian R. Shiels.

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
