## [Decision Letter · Decision Letter 0]

28 Sep 2021

PONE-D-21-24458Susceptibility to disease (tropical theileriosis) is associated with differential expression of host genes that possess motifs recognised by a pathogen DNA binding proteinPLOS ONE

Dear Brian,

Thank you for submitting your manuscript to PLOS ONE. After careful consideration, we feel that it has merit but does not fully meet PLOS ONE’s publication criteria as it currently stands. Therefore, we invite you to submit a revised version of the manuscript that addresses the points raised during the review process.

 First, I'd like to apologize fort he time taken to come to a decision. The first two reviewers submitted their quite favorable reports quickly, but the third reviewer's report has only just came in and you'll see it's quite critical of the mapping and pointed out that if your mapping strategy was misunderstood it's likely that it was poorly described. As handling editor I found also the PCA analysis and particularly its interpretation wasn't sufficiently explained. For example, there was some obvious outliers that were not mentioned and it was not stated whether the PCA analysis was the basis used to decrease to 5 from 6 the number of lines derived from B. taurus ; meaning that you ended up comparing 6 lines derived from B. indicus  to 5 lines from B. taurus and how this imbalance might have impacted on the results and your interpretation. So, I've marked the submission down for major revision to give you the opportunity to clarify better the strategies, techniques used and their consequences on the choices you made.

We look forward to receiving your revised manuscript.

Kind regards,

Gordon

Gordon Langsley

Academic Editor

PLOS ONE

Journal Requirements:

3. We note that you have stated that you will provide repository information for your data at acceptance. Should your manuscript be accepted for publication, we will hold it until you provide the relevant accession numbers or DOIs necessary to access your data. If you wish to make changes to your Data Availability statement, please describe these changes in your cover letter and we will update your Data Availability statement to reflect the information you provide

Reviewers' comments:

Reviewer's Responses to Questions

**Comments to the Author**

1. Is the manuscript technically sound, and do the data support the conclusions?

Reviewer #1: Yes

Reviewer #2: Yes

Reviewer #3: No

2. Has the statistical analysis been performed appropriately and rigorously? 

Reviewer #1: Yes

Reviewer #2: Yes

Reviewer #3: Yes

3. Have the authors made all data underlying the findings in their manuscript fully available?

Reviewer #1: Yes

Reviewer #2: No

Reviewer #3: Yes

4. Is the manuscript presented in an intelligible fashion and written in standard English?

Reviewer #1: Yes

Reviewer #2: Yes

Reviewer #3: Yes

5. Review Comments to the Author

Reviewer #1: The manuscript by Larcombe et al. studies an important topic in Theileria research. Identification of host/parasite associated mechanisms of cattle resistance/tolerance to tropical theileriosis can provide novel control strategies for theileriosis. Both resistant and susceptible animals get infected by Theileria sporozoites but the disease pathology is mild in resistant Sahiwals and Holsteins die of infection. Therefore, comparisons between these two groups of animals could provide new insights into Theileria annulata infection biology and disease.

The authors approach was to perform dual RNA-seq on low passages of Theileria annulata transformed cell lines established ex vivo from theileriosis susceptible (Bos taurus, Holstein(Ho)) and resistant (Bos indicus, Sahiwal (Sa)) cattle and looked at host and parasite genes differentially expressed between individuals of each group. Although the paper is descriptive and fully based on bioinformatics analyses, I found the findings on differential binding of TashAT2 to DNA in Holstein and Sahiwal interesting and the results could be useful for further development of projects on TashAT family as candidate host cell manipulator Theileria annulata genes.

Here are my specific comments on the manuscript:

-In figure 1 some animal samples show great variance and does not cluster with others, I’m aware that these are from individual animals. It would be if the authors discuss the possible reasons behind this variance.

Moreover in the same figure we can observe 12 points representing 12 samples (6 Ho versus 6 Sa), but in lines 157-158 (in fig1 legend as well) it is mentioned that 11 cell lines (5 cell lines of Holstein and 6 Sahiwal cattle) was studied for the PCA plot. Either there is an error in the figure or in the number of cell lines studied.

-In figure 1 legend Taurus should be taurus.

-Concerning differential expression of parasite genes between Sa and Ho, I was wondering if the authors have checked by western blot the protein levels of TashAT2 in nuclear extracts of Sa and Ho cell lines? Although TashAT2 mRNA levels was found not to be differentially expressed, the protein levels could be different (due to stabilization and etc.) More availability of TashAT2 in the host cell nucleus could result in occupation of more sites on DNA possibly leading to changes in gene expression.

-In the list of DE parasite genes (S10_File) between Sa and Ho there are several other interesting candidates that are not discussed or mentioned in results section. For example, we know that some of these DE genes (such as SVSPs) have nuclear localization signals and are found to be trafficked to the nucleus and contribute to cell transformation. Specific examples from this list are TA09790 (up in taurus) and TA05560 (up in indicus). This notion gets more importance when we consider T. parva infection. We know that resistance to T. parva have been reported in endemic African breeds cattle breeds and we know that the TashAT family have orthologues in T. parva but Tpsh family in parva lack DNA binding domains, so this means that other DNA-binding parasite proteins (SVSPs?) could also be involved in differential gene expression observed in disease resistant and disease susceptible animals.

-I think in the discussion authors can add few sentences concerning future possibilities to study their findings in more details such as comparative ChIP-sep/ChIP-qPCR approaches using anti-TashAT2 antibodies in Ho and Sa derived Theileria transformed cell lines. They might have already tested this.

-Lines 277, 278, 279: 1X, 2X and 3X instead of X1, X2 and X3

-I think mentioning COVID-19 several times in the manuscript is not necessary.

-It’s better to mention gene IDs of TashATs where they are introduced in the text for the first time.

- In lines 227-246 it is not clear whether these analyses were done on host cell genes or on parasite genes. I can imagine it’s the host cell genes?

Reviewer #2: In this manuscript, the authors perform RNAseq of Theileria infected cell lines to investigate disease susceptibility of tropical theileriosis. The comparative transcriptome analysis between tolerant (B. indicus, Sahiwal) and susceptible (B. taurus, Holstein) to acute disease identified differential transcriptome signatures whose pathways linked to innate immunity, cholesterol biosynthesis, cancer, and parasite infection. Differentially expressed genes (DEGs) of this comparison partially overlapped previously reported DEGs induced by Theileria infection in the BL20 cell line. Although conclusive parasite factors that cause different disease susceptibilities were not newly identified, the authors found binding motifs of TashAT2 are divergent in the two breeds genomes. In summary, this manuscript is suitable for publication in the journal. However, I have a few requests for corrections before publication.

Minor comments

Some text is unreadable due to the low resolution, especially figure 3, 4, 7, could you improve it?

The sizes of Table 1, 2, 3 are large and I couldn’t read all parts. These tables require modification.

Correct “Table S8” to “Table S6” of the legend in S6 Table.

Line 61, 1182, 1187, 1196, 1248, 1270: Correct “Vs” to “vs”.

Line 672: Correct “susceptible breeds” to “susceptible” breeds.

Reviewer #3: Susceptibility to disease (tropical theileriosis) is associated with differential expression of host genes that possess motifs recognized by a pathogen DNA binding protein

Author: Shiels et al.

In this work, the authors aim to identify host factors associated with susceptibility to tropical theileriosis, a disease caused by Theileria annulata in livestock. To this end, the authors compare RNAseq data from T. annulata-infected cell lines from Bos taurus (a susceptible species/breed to T. annulata infection) with similar data from infected cell lines from Bos indicus (a less susceptible species/breed). If I understand correctly from the methods, both RNAseq datasets were mapped to the Bos taurus genome and gene expression was tehn quantified and compared.

The work, as is, has a few key issues which, until addressed, make it hard to determine the significance of the work. In particular:

1. Bos indicus and Bos taurus are referred to as different breeds but their names indicate different species; how divergent are these taxa and how does that translate into genomic and transcriptomic differences?

2. Following from (1), the authors do not account for inherent gene expression differences between species;

3. Following from (1), the authors do not account for differences in mapping efficiency between species to the B. taurus genome. If, in fact, both RNAseq datasets were mapped to the B. taurus genome, it is likely that the data from B. taurus infected cell lines have a mapping success rate to the B. taurus genome that is higher than the reads from B. indicus.

4. If the two species differ significantly in gene expression (#2) or if mapping success differs between species (#3), either of those can, by itself, explain the pattern in Fig 1, as well as many of the differences observed between species for individual genes.

Species-specific differences in gene expression may be addressed by subtracting gene expression between infected and uninfected cells of the same species.

The impact of genome-wide differences can be addressed by:

- Demonstrating that there is no species-specific impact on read mapping success;

- If genomic differences do impact read mapping rate, one possible option is to map RNAseq data to each species’ genome and then merge the two expression quantification datasets based on gene orthology. If this is what has been done, please explain.

A possibly simpler approach to the question may be to determine, for each host species, differential gene expression between uninfected and infected cell lines, and then compare the two sets of DEGs. The intersection will possibly be host response to infection, which the unique responses in each host may reflect differences in susceptibility.

6. PLOS authors have the option to publish the peer review history of their article (what does this mean?). If published, this will include your full peer review and any attached files.

Reviewer #1: **Yes: **Shahin Tajeri

Reviewer #2: No

Reviewer #3: No

---

## [Author Response · Author response to Decision Letter 0]

12 Nov 2021

PONE-D-21-24458

Susceptibility to disease (tropical theileriosis) is associated with differential expression of host genes that possess motifs recognised by a pathogen DNA binding protein

PLOS ONE

Response to Editor and reviewers

We address the points made by you (the editor) and reviewers below and highlight the position in the manuscript where revision has been made. Altered text in manuscript highlighted in yellow.

Editor - the PCA analysis and particularly its interpretation wasn't sufficiently explained. For example, there was some obvious outliers that were not mentioned and it was not stated whether the PCA analysis was the basis used to decrease to 5 from 6 the number of lines derived from B. taurus; meaning that you ended up comparing 6 lines derived from B. indicus to 5 lines from B. taurus and how this imbalance might have impacted on the results and your interpretation. So, I've marked the submission down for major revision to give you the opportunity to clarify better the strategies, techniques used and their consequences on the choices you made.

Response

As indicated by Reviewer 1, this is a misunderstanding of the data based on an error in Figure 1, for which we apologise. There were only 11, low passage archived Sahiwal and Holstein infected cell lines available at the Roslin Institute and so only 11 samples of RNA-seq data generated. The error occurred when generating a high-quality figure of PCA for publication, a double click on the mouse apparently, and the error was not noticed in the submitted figure. Thus, there was never removal of an outlier to improve data comparison. A new corrected Figure 1 has been generated. Regarding whether having an unequal number of samples impacted on the analysis - this is unlikely. All it would do is alter the potential level of significance by generating a smaller sample set for one condition vs the other. It is not necessary to have two sample sets of exactly the same size to make a comparison, and all the differences identified between the breed sample sets meet a recognized statistical cut-off.

Regarding outliers and within sample (breed) set variability (also raised by Reviewer 1). It is clear from the PCA that there is good separation between samples representing the breeds but that there is also variability of samples within breed. Looking at the expression values across samples (File S2) it is easy to see why this is detected by the PCA; for some genes there is quite a lot of variance within a breed. Also based on published data i.e. PCR results across Holstein and Sahiwal samples for cytokines and other genes tested, a high level of within breed variance has been detected, previously (see refs 13, 21, 42), so within breed variance was expected. I have added some discussion (line 688-704) on PCA result and how within breed variability might occur. We agree that it is relevant that the variance should be highlighted and propose that each individual animal may show a unique response to infection. Despite the variance, our study does show a considerable number of genes that are statistically significant between the breeds, and a large number where differences are consistent across all samples for each breed. Based on the known high degree of variability in gene expression between infected cell lines (see also ref 65), this result was a pleasant surprise.

Reviewers' comments:

Reviewer #1: The manuscript by Larcombe et al. studies an important topic in Theileria research. Identification of host/parasite associated mechanisms of cattle resistance/tolerance to tropical theileriosis can provide novel control strategies for theileriosis. Both resistant and susceptible animals get infected by Theileria sporozoites but the disease pathology is mild in resistant Sahiwals and Holsteins die of infection. Therefore, comparisons between these two groups of animals could provide new insights into Theileria annulata infection biology and disease.

The authors approach was to perform dual RNA-seq on low passages of Theileria annulata transformed cell lines established ex vivo from theileriosis susceptible (Bos taurus, Holstein(Ho)) and resistant (Bos indicus, Sahiwal (Sa)) cattle and looked at host and parasite genes differentially expressed between individuals of each group. Although the paper is descriptive and fully based on bioinformatics analyses, I found the findings on differential binding of TashAT2 to DNA in Holstein and Sahiwal interesting and the results could be useful for further development of projects on TashAT family as candidate host cell manipulator Theileria annulata genes.

Response: We thank Dr. Tajeri for his insightful comments on the manuscript.

Specific comments on the manuscript:

-In figure 1 some animal samples show great variance and does not cluster with others, I’m aware that these are from individual animals. It would be if the authors discuss the possible reasons behind this variance.

Moreover, in the same figure we can observe 12 points representing 12 samples (6 Ho versus 6 Sa), but in lines 157-158 (in fig1 legend as well) it is mentioned that 11 cell lines (5 cell lines of Holstein and 6 Sahiwal cattle) was studied for the PCA plot. Either there is an error in the figure or in the number of cell lines studied.

See response to editor above 

-In figure 1 legend Taurus should be taurus. – changed to Holstein and Sahiwal 

-Concerning differential expression of parasite genes between Sahiwal and Holstein, I was wondering if the authors have checked by western blot the protein levels of TashAT2 in nuclear extracts of Sa and Ho cell lines? Although TashAT2 mRNA levels was found not to be differentially expressed, the protein levels could be different (due to stabilization and etc.) More availability of TashAT2 in the host cell nucleus could result in occupation of more sites on DNA possibly leading to changes in gene expression.

Response: We did not follow up on levels of host nuclear TashAT2 initially, because it was not highlighted in the parasite RNA-seq data and previous work indicated protein levels follow RNA. Nevertheless, the point is relevant and of interest. To investigate this, we conducted immunofluorescence using an antiserum specific to TashAT2. The results demonstrate that a) all cell lines express TashAT2 and b) there is no clear difference in host nuclear expression level between the three Holstein and Sahiwal lines that were tested (same lines used as for TA9 blot). This data has been added to the manuscript: line 608 to 611 and File S13. 

-In the list of DE parasite genes (S10_File) between Sa and Ho there are several other interesting candidates that are not discussed or mentioned in results section. For example, we know that some of these DE genes (such as SVSPs) have nuclear localization signals and are found to be trafficked to the nucleus and contribute to cell transformation. Specific examples from this list are TA09790 (up in taurus) and TA05560 (up in indicus). This notion gets more importance when we consider T. parva infection. We know that resistance to T. parva have been reported in endemic African breeds cattle breeds and we know that the TashAT family have orthologues in T. parva but Tpsh family in parva lack DNA binding domains, so this means that other DNA-binding parasite proteins (SVSPs?) could also be involved in differential gene expression observed in disease resistant and disease susceptible animals.

Response: The general result from our analysis is that there is not much evidence for large differences in parasite gene expression between the two breeds, and we analysed the best candidate (TA9) at the protein level. Re the two SVSPs: both show a less than 2-fold change (absolute). There are no reagents that have conclusively detected endogenous SVSPs by immunoblot, while the published IFAT (Schmuckli et al., Plos One 2009) only indicated a parasite location (nuclear localisation was by transfection into an uninfected cell line alone), there is no evidence SVSPs bind DNA and data suggests they may be highly diverse, rapidly degraded and are under no selective pressure (Weir et al., BMC genomics 2010). So personally, I do not think they are good candidates for follow up. Re the T. parva orthologues of TashAT family genes: It is unknown whether they lack DNA binding capability; from the genome sequences they lack a predicted canonical AT hook motif (RGRP core). However, several have predicted nuclear localisation sites with similarity to the T. annulata orthologue, and serum against TashAT2 detects the host nucleus in T. parva (Swan et al., Mol. Biol. Parasitol. 1999). The best current prediction is that T. parva orthologues go to the host nucleus and bind DNA but to a divergent sequence, probably linked to divergent evolution of the parasites in a different host cell lineage. 

-I think in the discussion authors can add few sentences concerning future possibilities to study their findings in more details such as comparative ChIP-sep/ChIP-qPCR approaches using anti-TashAT2 antibodies in Ho and Sa derived Theileria transformed cell lines. They might have already tested this.

I have added a line on requirement for ChIp in future studies (line 854). 

-Lines 277, 278, 279: 1X, 2X and 3X instead of X1, X2 and X3 – altered as requested.

-I think mentioning COVID-19 several times in the manuscript is not necessary.

Removed reference to Covid 19 at line 105, line 752 and 775.

-It’s better to mention gene IDs of TashATs where they are introduced in the text for the first time. Gene ID for TashAT2 given on line 42.

- In lines 227-246 it is not clear whether these analyses were done on host cell genes or on parasite genes. I can imagine it’s the host cell genes?

Response: Indeed, this was host cell genes – I have added infected host cell to the section title and bovine to the first sentence, for clarification. The rationale for this is that differences in what the infected cell secretes or transduces via receptors could have a major impact on differential interaction with the immune response.

Reviewer #2: In this manuscript, the authors perform RNAseq of Theileria infected cell lines to investigate disease susceptibility of tropical theileriosis. The comparative transcriptome analysis between tolerant (B. indicus, Sahiwal) and susceptible (B. taurus, Holstein) to acute disease identified differential transcriptome signatures whose pathways linked to innate immunity, cholesterol biosynthesis, cancer, and parasite infection. Differentially expressed genes (DEGs) of this comparison partially overlapped previously reported DEGs induced by Theileria infection in the BL20 cell line. Although conclusive parasite factors that cause different disease susceptibilities were not newly identified, the authors found binding motifs of TashAT2 are divergent in the two breeds genomes. In summary, this manuscript is suitable for publication in the journal. However, I have a few requests for corrections before publication.

Minor comments

Some text is unreadable due to the low resolution, especially figure 3, 4, 7, could you improve it?

Response: We hope that this is due to the build of the PDF on the web site, as we generated high quality images for each figure following journal guidelines. If this is not the case and figures need improved, we will do so prior to publication.

The sizes of Table 1, 2, 3 are large and I couldn’t read all parts. These tables require modification.

Response: We agree that Tables are large, and I have attempted to trim where possible. This however was minor, because if I had deleted further, the tables would not have matched the text and context of results would have been lost. Also, it might have looked like cherry picking most appropriate pathways. One option would be to move one or two Tables to supplementary (say Table 2), but I did not do this because it makes it harder for the reader to check what is said in the text. If required, Tables can be reduced with a change in text or moved to supplementary. Ditto if there is an issue with Tables being out with margins on manuscript – submission website indicated this was allowed.

Correct “Table S8” to “Table S6” of the legend in S6 Table.

Corrected

Line 61, 1182, 1187, 1196, 1248, 1270: Correct “Vs” to “vs”.

Corrected

Line 672: Correct “susceptible breeds” to “susceptible” breeds.

Corrected – now in edited section on PCA variability line 704 - 706

Reviewer #3: Susceptibility to disease (tropical theileriosis) is associated with differential expression of host genes that possess motifs recognized by a pathogen DNA binding protein

Author: Shiels et al.

In this work, the authors aim to identify host factors associated with susceptibility to tropical theileriosis, a disease caused by Theileria annulata in livestock. To this end, the authors compare RNAseq data from T. annulata-infected cell lines from Bos taurus (a susceptible species/breed to T. annulata infection) with similar data from infected cell lines from Bos indicus (a less susceptible species/breed). If I understand correctly from the methods, both RNAseq datasets were mapped to the Bos taurus genome and gene expression was tehn quantified and compared.

The work, as is, has a few key issues which, until addressed, make it hard to determine the significance of the work. In particular:

1. Bos indicus and Bos taurus are referred to as different breeds but their names indicate different species; how divergent are these taxa and how does that translate into genomic and transcriptomic differences?

2. Following from (1), the authors do not account for inherent gene expression differences between species;

3. Following from (1), the authors do not account for differences in mapping efficiency between species to the B. taurus genome. If, in fact, both RNAseq datasets were mapped to the B. taurus genome, it is likely that the data from B. taurus infected cell lines have a mapping success rate to the B. taurus genome that is higher than the reads from B. indicus.

4. If the two species differ significantly in gene expression (#2) or if mapping success differs between species (#3), either of those can, by itself, explain the pattern in Fig 1, as well as many of the differences observed between species for individual genes.

Species-specific differences in gene expression may be addressed by subtracting gene expression between infected and uninfected cells of the same species.

The impact of genome-wide differences can be addressed by:

- Demonstrating that there is no species-specific impact on read mapping success;

- If genomic differences do impact read mapping rate, one possible option is to map RNAseq data to each species’ genome and then merge the two expression quantification datasets based on gene orthology. If this is what has been done, please explain.

A possibly simpler approach to the question may be to determine, for each host species, differential gene expression between uninfected and infected cell lines, and then compare the two sets of DEGs. The intersection will possibly be host response to infection, which the unique responses in each host may reflect differences in susceptibility.

Response: This reviewer suggests that we should have mapped to each genome separately as the strategy of mapping RNA-seq reads from infected cells from both sub-species to Bos taurus may have resulted in a bias in detection of differential gene expression (i.e. difference in mapping success rate/higher for Bos taurus). We do not believe this has occurred or that it was necessary based on the following. 

The common ancestor of indicine and taurine cattle is the auroch, an ancient species of wild cattle. Indicine and taurine cattle, being very closely related, can easily inter-breed; this ability is, in fact, the basis of a variety breed improvement to combat diseases such as theileriosis. The taxonomy of these cattle has been the subject of much debate; as they are so close, they have been classified as sister sub-species. The utility of mapping the RNA-seq data separately onto both the Bos indicus and Bos taurus genomes was considered but was rejected on the basis of the fact that the assembly and annotation of the Bos taurus reference genome is far superior. The quality of read mapping would not have been equivalent and had we undertaken this approach, we would have created two disparate datasets exhibiting inherent, unacceptable bias when combined. For this reason, it was decided to do a two-way comparison based on the better, quality reference genome, in the likelihood that there would be little differences in the read mapping of the two cattle types based on identity of mRNA coding regions, reflecting the approach of previous work in the field (ref 23). 

The reviewer indicated that “The impact of genome-wide differences can be addressed by…demonstrating that there is no species-specific impact on read mapping success”. 

Response: We performed this analysis, which confirms equitable mapping success and that there is no effect of cattle breed type. This analysis and data presented in the manuscript demonstrate that:

(a) The % reads mapped for each Holstein (Bos taurus) sample was equivalent to that for each Sahiwal (Bos indicus) sample and there is no evidence of any group-related difference. We thank the reviewer for this suggestion and add the information to the manuscript as Table S3.

(b) Nearly half the differentially expressed genes in the dataset display elevated expression in the Sahiwal group.

(c) Every gene that was selected for qRT-PCR including, those lower in Sahiwal group, reflected the RNA seq result, and confirmed that within-breed variability in expression (even across half sibs) was genuine. These results essentially repeat the above-mentioned microarray study where qPCR was used to validate differential hybridisation results. So, to date, qPCR has not highlighted any erroneous results in high-throughput gene expression analysis based on our approach.

(d) Comparison of the TashAT2 motif dataset across the two genomes with the subset of differentially expressed genes showed considerable overlap in pathways known to be altered by infection e.g. PI3K and ILK signalling and included genes expressed at a lower level in Sahiwal (e.g. SLIT, PRUNE, ITGs) and altered by infection. Taken together, our results support a lack of bias in the DE results and identification of infection-associated DE between the breeds linked to TashAT2 motifs. 

Re the suggestion of generating a DE dataset for both Hosltein and Sahiwal lines, we agree that in the future this could/should be performed for multiple infected and uninfected lines/cells to further investigate within- and between-breed variability and so have stated this in the discussion: line 702.

2. Following from (1), the authors do not account for inherent gene expression differences between species;

Response: We present a significant subset of genes that are altered by infection in a Bos taurus cell line (Fig. 2), with a model that has been utilised to detect infection associated DE in most studies over the last 30 years. We highlighted that this was performed to identify DE linked to infection and distinguish these genes from those displaying possible inherent expression differences between the two breeds. Our data shows remarkable overlap between the top pathways enriched in both the full and the infection-associated DE datasets, many of which have been linked to infection/transformation of bovine leukocytes by the Theileria parasite. Moreover, we do not think that inherent differences between breeds should be totally discounted because, at the end of the day, it is the difference between the gene expression profile of the infected cell that will determine its phenotype/virulence, particularly genes with links to the immune response (e.g. inducible ISGs) and cancer highlighted in the paper. These could arise from a change in gene expression caused by infection of cells in one breed but not the other, DE linked to infection in both breeds, or a difference not altered by infection (inherently different). Thus, it is logical, reasonable and informative to directly compare infected cells from both types of cattle for all differentially expressed genes, in addition to those generated by parasite infection of the cell.

In summary, we believe finding overlaps between genes differentially expressed between the two breeds, infected and uninfected cells and bearing TashAT2 motifs will be of interest to PLOS One readers. We hope you agree that our revisions and response to reviewers now meet the requirements of the Journal.

---

## [Decision Letter · Decision Letter 1]

2 Dec 2021

PONE-D-21-24458R1

Susceptibility to disease (tropical theileriosis) is associated with differential expression of host genes that possess motifs recognised by a pathogen DNA binding protein

PLOS ONE

Dear Brian,

Thank you for submitting your manuscript to PLOS ONE. After careful consideration, we feel that it has merit but does not fully meet PLOS ONE’s publication criteria as it currently stands. Therefore, we invite you to submit a revised version of the manuscript that addresses the points raised during the review process.

As you know there is no proof reading at PLoS One, and as referee #3 requested a limited number of small corrections to the text, I have marked your submission down for minor revision to give you the opportunity to make the minor text changes.

We look forward to receiving your revised manuscript.

Kind regards,

Gordon

Gordon Langsley

Academic Editor

PLOS ONE

Reviewers' comments:

Reviewer's Responses to Questions

**Comments to the Author**

1. If the authors have adequately addressed your comments raised in a previous round of review and you feel that this manuscript is now acceptable for publication, you may indicate that here to bypass the “Comments to the Author” section, enter your conflict of interest statement in the “Confidential to Editor” section, and submit your "Accept" recommendation.

Reviewer #1: All comments have been addressed

Reviewer #2: All comments have been addressed

Reviewer #3: All comments have been addressed

2. Is the manuscript technically sound, and do the data support the conclusions?

Reviewer #1: Yes

Reviewer #2: Yes

Reviewer #3: Yes

3. Has the statistical analysis been performed appropriately and rigorously? 

Reviewer #1: Yes

Reviewer #2: Yes

Reviewer #3: Yes

4. Have the authors made all data underlying the findings in their manuscript fully available?

Reviewer #1: Yes

Reviewer #2: Yes

Reviewer #3: Yes

5. Is the manuscript presented in an intelligible fashion and written in standard English?

Reviewer #1: Yes

Reviewer #2: Yes

Reviewer #3: Yes

6. Review Comments to the Author

Reviewer #1: (No Response)

Reviewer #2: The authors have responded all requested correction and the manuscript has been improved. I think this manuscript is acceptable now.

Reviewer #3: Susceptibility to disease (tropical theileriosis) is associated with differential expression of host genes that possess motifs recognized by a pathogen DNA binding protein

Author: Shiels et al.

This is a resubmission of a paper in which authors compare patterns of gene expression in ex vivo Theileria annulata-infected bovine cells lines from cattle breeds susceptible or tolerant to T. annulata infection to identify host and parasite factors that impact the susceptibility phenotype. The authors found plausible links between differences in tolerance and interferon-stimulated gene pathways, production of INFB1, inflammatory responses and oncogenesis. Critically, they identify a motif bound by a parasite gene that is differently distributed in the genome of both breeds. The authors find that genes with differences between breeds in the presence/absence/number of these motifs were enriched for functions potentially connected to infection.

This is a well-written paper with several interesting findings, leading to new hypotheses related to interactions between host and parasite, as well as infection susceptibility.

This resubmission addresses my previous concerns adequately.

Some minor issues:

- Pls add a first line to all supplemental word and xls tables with the title of the table

- Lack of consistency throughout text in use of versus, vs, vs. (best to use an italicized form, as vs.)

- L136, word missing in “offer the best available for“

- L191: change “mapped to the bovine” to “mapped to the Bos taurus”

- L319: change “number of reads” to “proportion of reads” (also, see typo in table S3)

- L629-630” “Subsequent PANTHER analysis showed”. Which genes were fed to this analysis? All 11,983 from B. taurus? Only the 7,575 shared? Those not shared? ...? The interpretation of the enrichment analysis will be affected.

- Fig 7C. I suggest using in Panel C different colors from those in legend in panel B

- L707-708: “This result may indicate that a number of infection-associated genes differentially expressed between breeds have been missed by our analysis “ I am not sure I follow. Couldn’t there be constitutively expressed genes, differentially expressed between breeds, regardless of infection status?

7. PLOS authors have the option to publish the peer review history of their article (what does this mean?). If published, this will include your full peer review and any attached files.

Reviewer #1: No

Reviewer #2: No

Reviewer #3: No

---

## [Author Response · Author response to Decision Letter 1]

10 Dec 2021

PONE-D-21-24458R1

Susceptibility to disease (tropical theileriosis) is associated with differential expression of host genes that possess motifs recognised by a pathogen DNA binding protein

PLOS ONE

Response to reviewers,

Thanks to the reviewers for making the effort to look at our manuscript for a second time. We were pleased that the three reviewers indicated that we had adequately answered all previous concerns. 

In response to minor issues raised by Reviewer 3, we have made the following amendments/corrections 

Reviewer 3.

This resubmission addresses my previous concerns adequately.

Some minor issues:

- Pls add a first line to all supplemental word and xls tables with the title of the table

- Response – edited as requested

- Lack of consistency throughout text in use of versus, vs, vs. (best to use an italicized form, as vs.)

- Response - corrected to vs. throughout

- L136, word missing in “offer the best available for“

- Response – edited to “offer the best infected cell lines available….”

- L191: change “mapped to the bovine” to “mapped to the Bos taurus” 

- Response – corrected to “mapped to the B. taurus 

- L319: change “number of reads” to “proportion of reads” (also, see typo in table S3)

- Response – change as requested

- L629-630” “Subsequent PANTHER analysis showed”. Which genes were fed to this analysis? All 11,983 from B. taurus? Only the 7,575 shared? Those not shared? ...? The interpretation of the enrichment analysis will be affected.

- Response – edited to “Subsequent PANTHER analysis of the shared genes data set showed an enrichment”

- Fig 7C. I suggest using in Panel C different colors from those in legend in panel B

- Response - colours in panel C changed to Yellow and Grey

- L707-708: “This result may indicate that a number of infection-associated genes differentially expressed between breeds have been missed by our analysis “ I am not sure I follow. Couldn’t there be constitutively expressed genes, differentially expressed between breeds, regardless of infection status?

- We agree the logic of the text preceding this statement (i.e. that the pathways identified for both data sets are more or less the same) could be confusing, and the reviewer is correct that non-infection associated genes could be/are differentially expressed between breeds. Nevertheless, published data indicates we have missed candidates, so the text has been edited with “This result may indicate” deleted. The text now reads – “However, previous studies indicate that a number of infection-associated genes differentially expressed between breeds have been missed by our analysis, and there are at least several candidates.”

---

## [Editor Report · Decision Letter 2]

16 Dec 2021

Susceptibility to disease (tropical theileriosis) is associated with differential expression of host genes that possess motifs recognised by a pathogen DNA binding protein

PONE-D-21-24458R2

Dear Brian,

We’re pleased to inform you that your manuscript has been judged scientifically suitable for publication and will be formally accepted for publication once it meets all outstanding technical requirements.

Kind regards,

Gordon

Gordon Langsley

Academic Editor

PLOS ONE
---

## [Editor Report · Acceptance letter]

20 Dec 2021

PONE-D-21-24458R2 

Susceptibility to disease (tropical theileriosis) is associated with differential expression of host genes that possess motifs recognised by a pathogen DNA binding protein 

Dear Dr. Shiels:

I'm pleased to inform you that your manuscript has been deemed suitable for publication in PLOS ONE. Congratulations! Your manuscript is now with our production department. 

Kind regards, 

on behalf of

Dr. Gordon Langsley 

Academic Editor

PLOS ONE